# ARTree: A Deep Autoregressive Model for Phylogenetic Inference

**Tianyu Xie**[†], **Cheng Zhang**[†,‡,*]

[†] School of Mathematical Sciences, Peking University
[‡] Center for Statistical Science, Peking University
tianyuxie@pku.edu.cn, chengzhang@math.pku.edu.cn

## Abstract

Designing flexible probabilistic models over tree topologies is important for developing efficient phylogenetic inference methods. To do that, previous works often leverage the similarity of tree topologies via hand-engineered heuristic features which would require pre-sampled tree topologies and may suffer from limited approximation capability. In this paper, we propose a deep autoregressive model for phylogenetic inference based on graph neural networks (GNNs), called ARTree. By decomposing a tree topology into a sequence of leaf node addition operations and modeling the involved conditional distributions based on learnable topological features via GNNs, ARTree can provide a rich family of distributions over the entire tree topology space that have simple sampling algorithms and density estimation procedures, without using heuristic features. We demonstrate the effectiveness and efficiency of our method on a benchmark of challenging real data tree topology density estimation and variational Bayesian phylogenetic inference problems.

## 1   Introduction

Reconstructing the evolutionary relationships among species has been one of the central problems in computational biology, with a wide range of applications such as genomic epidemiology (Dudas et al., 2017; du Plessis et al., 2021; Attwood et al., 2022) and conservation genetics (DeSalle & Amato, 2004). Based on molecular sequence data (e.g. DNA, RNA, or protein sequences) of the observed species and a model of evolution, this has been formulated as a statistical inference problem on the hypotheses of shared history, i.e., *phylogenetic trees*, where maximum likelihood and Bayesian approaches are the most popular methods (Felsenstein, 1981; Yang & Rannala, 1997; Mau et al., 1999; Larget & Simon, 1999; Huelsenbeck et al., 2001). However, phylogenetic inference can be challenging due to the composite structure of tree space which contains both continuous and discrete components (e.g., the branch lengths and the tree topologies) and the large search space of tree topologies that explodes combinatorially as the number of species increases (Whidden & Matsen IV, 2015; Dinh et al., 2017).

Recently, several efforts have been made to improve the efficiency of phylogenetic inference algorithms by designing flexible probabilistic models over the tree topology space (Höhna & Drummond, 2012; Larget, 2013; Zhang & Matsen IV, 2018). One typical example is subsplit Bayesian networks (SBNs) (Zhang & Matsen IV, 2018), which is a powerful probabilistic graphical model that provides a flexible family of distributions over tree topologies. Given a sample of tree topologies (e.g., sampled tree topologies from an MCMC run), SBNs have proved effective for accurate tree topology density estimation that generalizes beyond observed samples by leveraging the similarity of hand-engineered subsplit structures among tree topologies. Moreover, SBNs also allow fast ancestral sampling and

---

[*]Corresponding author.

37th Conference on Neural Information Processing Systems (NeurIPS 2023).

hence were later on integrated into a variational Bayesian phylogenetic inference (VBPI) framework to provide variational posteriors over tree topologies (Zhang & Matsen IV, 2019). However, due to the limited parent-child subsplit patterns in the observed samples, SBNs can not provide distributions whose support spans the entire tree topology space (Zhang & Matsen IV, 2022). Furthermore, when used as variational distributions over tree topologies in VBPI, SBNs often rely on subsplit support estimation for variational parameterization, which requires high-quality pre-sampled tree topologies that would become challenging to obtain when the posterior is diffuse.

While SBNs suffer from the aforementioned limitations due to their hand-engineered design, a number of deep learning methods have been proposed for probabilistic modeling of graphs (Jin et al., 2018; You et al., 2018a; Cao & Kipf, 2018; Simonovsky & Komodakis, 2018). Instead of using hand-engineered features, these approaches use neural networks to define probabilistic models for the connections between graph nodes which allow for learnable distributions over graphs. Due to the flexibility of neural networks, the resulting models are capable of learning complex graph patterns automatically. Among these deep graph models, graph autoregressive models (You et al., 2018b; Li et al., 2018; Liao et al., 2019; Dai et al., 2020; Shi et al., 2020) are designed to learn flexible graph distributions that also allow easy sampling procedures by sequentially adding nodes and edges. Therefore, they serve as an ideal substitution of SBNs for phylogenetic inference that can provide more expressive distributions over tree topologies.

In this paper, we propose a novel deep autoregressive model for phylogenetic inference, called ARTree, which allows for more flexible distributions over tree topologies without using heuristic features than SBNs. With a pre-selected order of leaf nodes (i.e., species or taxa), ARTree generates a tree topology by recursively adding new leaf nodes to the edges of the current tree topology, starting from a star-shaped tree topology with the first three leaf nodes (Figure 1). The edge to which a new leaf node connects is determined according to a conditional distribution based on learnable topological features of the current tree topology via GNNs (Zhang, 2023). This way, probability distributions provided by ARTree all have full support that spans the entire tree topology space. Unlike SBNs, ARTree can be readily used in VBPI without requiring subsplit support estimation for parameterization. In experiments, we show that ARTree outperforms SBNs on a benchmark of challenging real data tree topology density estimation and variational Bayesian phylogenetic inference problems.

## 2 Background

**Phylogenetic likelihoods** A phylogenetic tree is commonly described by a bifurcating tree topology $\tau$ and the associated non-negative branch lengths $\boldsymbol{q}$. The tree topology $\tau$ represents the evolutionary relationship of the species and the branch lengths $\boldsymbol{q}$ quantify the evolutionary intensity along the edges of $\tau$. The leaf nodes of $\tau$ correspond to the observed species and the internal nodes of $\tau$ represent the unobserved ancestor species. A continuous time Markov model is often used to describe the transition probabilities of the characters along the edges of the tree (Felsenstein, 2004). Concretely, let $\boldsymbol{Y} = \{Y_1, \ldots, Y_M\} \in \Omega^{N \times M}$ be the observed sequences (with characters in $\Omega$) of length $M$ over $N$ species. Under the assumption that different sites evolve independently and identically, the likelihood of $\boldsymbol{Y}$ given $\tau, \boldsymbol{q}$ takes the form

$$p(\boldsymbol{Y}|\tau, \boldsymbol{q}) = \prod_{i=1}^{M} p(Y_i|\tau, \boldsymbol{q}) = \prod_{i=1}^{M} \sum_{a^i} \eta(a_r^i) \prod_{(u,v) \in E(\tau)} P_{a_u^i a_v^i}(q_{uv}), \tag{1}$$

where $a^i$ ranges over all extensions of $Y_i$ to the internal nodes with $a_u^i$ being the character assignment of node $u$ ($r$ represents the root node), $E(\tau)$ is the set of edges of $\tau$, $q_{uv}$ is the branch length of the edge $(u, v) \in E(\tau)$, $P_{jk}(q)$ is the transition probability from character $j$ to $k$ through a branch of length $q$, and $\eta$ is the stationary distribution of the Markov model.

**Subsplit Bayesian networks** Let $\mathcal{X}$ be the set of leaf labels representing the existing species. A non-empty subset of $\mathcal{X}$ is called a *clade* and the set of all clades $\mathcal{C}(\mathcal{X})$ is equipped with a total order $\succ$ (e.g., lexicographical order). An ordered clade pair $(W, Z)$ satisfying $W \cap Z = \emptyset$ and $W \succ Z$ is called a *subsplit*. A *subsplit Bayesian network* (SBN) is then defined as a Bayesian network whose nodes take subsplit values or singleton clade values that describe the local topological structures of tree topologies. For a rooted tree topology, one can find the corresponding node assignment of SBNs

by following its splitting processes (Figure 4 in Appendix A). The SBN based probability of a rooted tree topology $\tau$ then takes the following form

$$p_{\text{sbn}}(T = \tau) = p(S_1 = s_1) \prod_{i>1} p(S_i = s_i | S_{\pi_i} = s_{\pi_i}), \tag{2}$$

where $S_i$ denotes the subsplit- or singleton-clade-valued random varaibles at node $i$ (node 1 is the root node), $\pi_i$ is the index set of the parents of node $i$ and $\{s_i\}_{i \geq 1}$ is the corresponding node assignment. For unrooted tree topologies, we can also define their SBN based probabilities by viewing them as rooted tree topologies with unobserved roots and integrating out the positions of the root node as follows: $p_{\text{sbn}}(T^{\text{u}} = \tau) = \sum_{e \in E(\tau)} p_{\text{sbn}}(\tau^e)$, where $\tau^e$ is the resulting rooted tree topology when the rooting position is on edge $e$. In practice, the conditional probability tables (CPTs) of SBNs are often parameterized based on a sample of tree topologies (e.g., the observed data for density estimation (Zhang & Matsen IV, 2018) or fast bootstrap/MCMC samples (Minh et al., 2013; Zhang, 2020) for VBPI). As a result, the supports of SBN-induced distributions are often limited by the splitting patterns in the observed samples and could not span the entire tree topology space (Zhang & Matsen IV, 2022). More details on SBNs can be found in Appendix A.

**Variational Bayesian phylogenetic inference**   Given a prior distribution $p(\tau, \boldsymbol{q})$, the phylogenetic posterior distribution takes the form

$$p(\tau, \boldsymbol{q}|\boldsymbol{Y}) = \frac{p(\boldsymbol{Y}|\tau, \boldsymbol{q})p(\tau, \boldsymbol{q})}{p(\boldsymbol{Y})} \propto p(\boldsymbol{Y}|\tau, \boldsymbol{q})p(\tau, \boldsymbol{q}). \tag{3}$$

Let $Q_{\boldsymbol{\phi}}(\tau)$ and $Q_{\boldsymbol{\psi}}(\boldsymbol{q}|\tau)$ be variational families over the spaces of tree topologies and branch lengths respectively. The VBPI approach uses $Q_{\boldsymbol{\phi}, \boldsymbol{\psi}}(\tau, \boldsymbol{q}) = Q_{\boldsymbol{\phi}}(\tau)Q_{\boldsymbol{\psi}}(\boldsymbol{q}|\tau)$ to approximate the posterior $p(\tau, \boldsymbol{q}|\boldsymbol{Y})$ by maximizing the following multi-sample ($K > 1$) lower bound

$$L^K(\boldsymbol{\phi}, \boldsymbol{\psi}) = \mathbb{E}_{\{(\tau^i, \boldsymbol{q}^i)\}_{i=1}^K \overset{\text{i.i.d.}}{\sim} Q_{\boldsymbol{\phi}, \boldsymbol{\psi}}} \log \left( \frac{1}{K} \sum_{i=1}^K \frac{p(\boldsymbol{Y}|\tau^i, \boldsymbol{q}^i)p(\tau^i, \boldsymbol{q}^i)}{Q_{\boldsymbol{\phi}}(\tau^i)Q_{\boldsymbol{\psi}}(\boldsymbol{q}^i|\tau^i)} \right). \tag{4}$$

The tree topology distribution $Q_{\boldsymbol{\phi}}(\tau)$ is often SBNs which in this case rely on subsplit support estimation for parameterization that requires high-quality pre-sampled tree topologies and would become challenging for diffuse posteriors (Zhang & Matsen IV, 2022). The branch lengths distribution $Q_{\boldsymbol{\psi}}(\boldsymbol{q}|\tau)$ can be diagonal lognormal distribution parametrized via heuristic features or learnable topological features of $\tau$ (Zhang & Matsen IV, 2019; Zhang, 2020, 2023). Compared to the single-sample lower bound, the multi-sample lower bound in (4) allows efficient variance-reduced stochastic gradient estimators (e.g. VIMCO (Mnih & Rezende, 2016)) for tree topology variational parameters. Moreover, using multiple samples would encourage exploration over the vast tree topology space, albeit it may also deteriorates the training of the variational approximation (Rainforth et al., 2019). In practice, a moderate $K$ is often suggested (Zhang & Matsen IV, 2022). See more details on VBPI in Appendix B.

**Graph autoregressive models**   By decomposing a graph as a sequence of components (nodes, edges, motifs, etc), graph autoregressive models generate the full graph by adding one component at a time, until some stopping criteria are satisfied (You et al., 2018b; Jin et al., 2018; Liao et al., 2019). In previous works, recurrent neural networks (RNNs) for graphs are usually utilized to predict new graph components conditioned on the sub-graphs generated so far (You et al., 2018b). The key of graph autoregressive models is to find a way to efficiently sequentialize graph structures, which is often domain-specific.

## 3   Proposed method

In this section, we propose ARTree, a deep autoregressive model for phylogenetic inference that can provide flexible distributions whose support spans the entire tree topology space and can be naturally parameterized without using heuristic approaches such as subsplit support estimation. We first describe a particular autoregressive generating process of phylogenetic tree topologies. We then develop powerful GNNs to parameterize learnable conditional distributions of this generating process. We consider unrooted tree topologies in this section, but the method developed here can be easily adapted to rooted tree topologies.

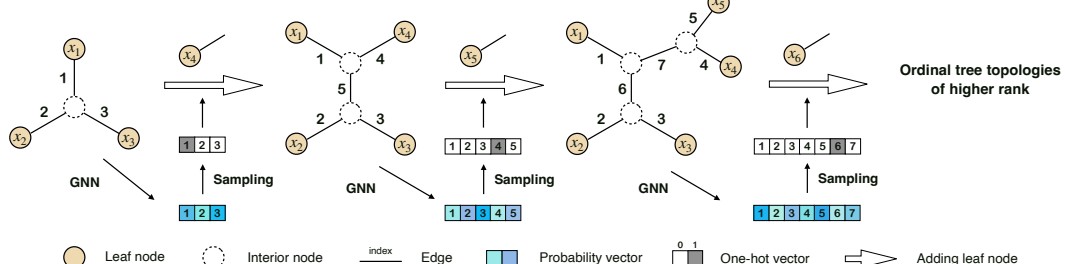

Figure 1: An overview of ARTree for autoregressive tree topology generation. The left plot is the starting ordinal tree topology of rank 3. This tree topology is then fed into GNNs which output a probability vector over edges. We then sample from the corresponding edge decision distribution and attach the next leaf node to the sampled edge. This process continues until an ordinal tree topology of rank $N$ is reached.

## 3.1 A sequential generating process of tree topologies

To better illustrate our approach, we begin with some notations. Let $\tau_n = (V_n, E_n)$ be a tree topology with $n$ leaf nodes and $V_n, E_n$ are the sets of nodes and edges respectively. Note that $|V_n| = 2n - 2$ and $|E_n| = 2n - 3$ due to the unrooted and bifurcating structure of $\tau_n$. The leaf nodes in $V_n$ are treated as labeled nodes and the interior nodes in $V_n$ are treated as unlabeled nodes. Let us assume a pre-selected order for the leaf nodes $\mathcal{X} = \{x_1, \ldots, x_N\}$, which is called taxa order for short by us. Now, consider a sequential generating process for all possible tree topologies that have leaf nodes $\mathcal{X}$. We start with a definition below.

**Definition 1** (Ordinal Tree Topology). *Let $\mathcal{X} = \{x_1, \ldots, x_N\}$ be a set of $N(N \geq 3)$ leaf nodes. Let $\tau_n = (V_n, E_n)$ be a tree topology with $n(n \leq N)$ leaf nodes in $\mathcal{X}$. We say $\tau_n$ is an ordinal tree topology of rank $n$, if its leaf nodes are the first $n$ elements of $\mathcal{X}$, i.e., $V_n \cap \mathcal{X} = \{x_1, \ldots, x_n\}$.*

We now describe a procedure that constructs ordinal tree topologies of rank $N$ recursively by adding one leaf node at a time as follows. We first start from $\tau_3$, the ordinal tree topology of rank 3, which is the smallest ordinal tree topology and is unique due to its unrooted and bifurcating structure. Suppose now we have an ordinal tree topology $\tau_n = (V_n, E_n)$ of rank $n$. To add the leaf node $x_{n+1}$ to $\tau_n$, we i) select an edge $e_n = (u, v) \in E_n$ and remove it from $E_n$; ii) add a new node $w$ and two new edges $(u, w), (w, v)$ to the tree topology; iii) add the leaf node $x_{n+1}$ and an edge $(w, x_{n+1})$ to the tree topology. This way, we obtain an ordinal tree topology $\tau_{n+1}$ of rank $n + 1$. Intuitively, the leaf node $x_{n+1}$ is added to the tree topology $\tau_n$ by attaching it to an existing edge $e_n \in E_n$. The position of the selected edge represents the evolutionary relationship between this new species and others. After performing this procedure for $n = 3, \ldots, N - 1$, we finally obtain an ordinal tree topology $\tau = \tau_N$ of rank $N$. See Figure 1 for an illustration.

During the generating process described above, the selected edges at each time step form a sequence $D = (e_3, \ldots, e_{N-1})$. This sequence $D$ of length $N - 3$ records all the decisions we have made for autoregressively generating a tree topology $\tau$ and thus we call $D$ a decision sequence. In fact, there is a one-to-one mapping between decision sequences and ordinal tree topologies of rank $N$, which is formalized in Theorem 1. Note that a similar process is also used in online phylogenetic sequential Monte Carlo (OPSMC) (Dinh et al., 2016), where the leaf node addition operation is incorporated into the design of the proposal distributions.

**Theorem 1.** *Let $\mathcal{D} = \{D | D = (e_3, \ldots, e_{N-1}), e_n \in E_n, \forall 3 \leq n \leq N - 1\}$ be the set of all decision sequences of length $N - 3$ and $\mathcal{T}$ be the set of all ordinal tree topologies of rank $N$. Let the map*

$$g: \begin{array}{rcl} \mathcal{D} & \to & \mathcal{T} \\ D & \mapsto & \tau \end{array}$$

*be the generating process described above. Then $g$ is a bijection between $\mathcal{D}$ and $\mathcal{T}$.*

According to Theorem 1, for each tree topology $\tau \in \mathcal{T}$, there is a unique decision sequence given by $g^{-1}(\tau)$. We call this process of finding the decision sequences of tree topologies the *decomposition*

---
**Algorithm 1:** ARTree: An autoregressive model for phylogenetic tree topologies

> **Input:** a set $\mathcal{X} = \{x_1, \ldots, x_N\}$ of leaf nodes.
> **Output:** an ordinal tree topology $\tau$ of rank $N$; the ARTree probability $Q(\tau)$ of $\tau$.
> $\tau_3 = (V_3, E_3) \leftarrow$ the unique ordinal tree topology of rank 3;
> **for** $n = 3, \ldots, N-1$ **do**
> > Calculate the probability vector $q_n \in \mathbb{R}^{|E_n|}$ using the current GNN model;
> > Sample an edge decision $e_n$ from Discrete $(q_n)$ and assume $e_n = (u, v)$;
> > Create a new node $w$;
> > $E_{n+1} \leftarrow (E_n \backslash \{e_n\}) \cup \{(u, w), (w, v), (w, x_{n+1})\}$;
> > $V_{n+1} \leftarrow V_n \cup \{w, x_{n+1}\}$;
> > $\tau_{n+1} \leftarrow (V_{n+1}, E_{n+1})$;
> **end**
> $\tau \leftarrow \tau_N$;
> $Q(\tau) \leftarrow q_3(e_3) q_4(e_4) \cdots q_{N-1}(e_{N-1})$.
---

*process.* See more details on the decomposition process in Appendix C. The following lemma shows that one can find $g^{-1}(\tau)$ in linear time.

**Lemma 1.** *The time complexity of the decomposition process induced by* $g^{-1}(\cdot)$ *is* $O(N)$.

The proofs of Theorem 1 and Lemma 1 can be found in Appendix D. Based on the bijection $g$ defined in Theorem 1, we can model the distribution $Q(D)$ over the space of decision sequences $\mathcal{D}$ instead of modeling the distribution $Q(\tau)$ over $\mathcal{T}$. Due to the sequential nature of $D$, we can decompose $Q(D)$ as the product of conditional distributions over the elements:

$$Q(D) = \prod_{n=3}^{N-1} Q(e_n | e_3, \ldots, e_{n-1}). \tag{5}$$

In what follows, we simplify $Q(e_n | e_3, \ldots, e_{n-1})$ as $Q(e_n | e_{<n})$ and let $e_{<3}$ be the empty set.

### 3.2 Graph neural networks for edge decision distribution

By Theorem 1, the sequence $e_{<n}$ corresponds to a sequence of ordinal tree topologies of increasing ranks $(\tau_3, \ldots, \tau_n)$ (the empty set $e_{<3}$ corresponds to the unique ordinal tree topology $\tau_3$ of rank 3). Therefore, the discrete distribution $Q(e_n | e_{<n})$ in equation (5) defines the probability of adding the leaf node $x_{n+1}$ to the edge $e_n$ of $\tau_n$, conditioned on all the ordinal tree topologies $(\tau_3, \ldots, \tau_n)$ generated so far. In what follows, we will show step by step how to use graph neural networks (GNNs) to parameterize such a conditional distribution given tree topologies.

**Topological node embeddings** At the $n$-th time step of the generating process, we first find the node embeddings of the current tree topology $\tau_n = (V_n, E_n)$, which is a set $\{f_n(u) \in \mathbb{R}^N : u \in V_n\}$ that assigns each node with an encoding vector in $\mathbb{R}^N$. Following Zhang (2023), we first assign one hot encoding to the leaf nodes, i.e.

$$[f_n(x_i)]_j = \delta_{ij}, 1 \le i \le n, 1 \le j \le N, \tag{6}$$

where $\delta$ is Kronecker delta function; we then get the embeddings for the interior nodes by minimizing the Dirichlet energy $\ell(f_n, \tau_n) := \sum_{(u,v) \in E_n} ||f_n(u) - f_n(v)||^2$ using the efficient two-pass algorithm described in Zhang (2023). One should note that the embeddings for interior nodes may change as new leaf nodes are added to the ordinal tree topologies, which is a main difference between our model and other graph autoregressive models.

**Message passing networks** Using these topological node embeddings as the initial node features, GNNs apply message passing steps to compute the representation vector of nodes that encode topological information of $\tau_n$, where the node features are updated with the information from their neighborhoods in a convolutional manner (Gilmer et al., 2017). More concretely, the $l$-th round of message passing is implemented by

$$m_n^l(u, v) = F_{\text{message}}^l(f_n^l(u), f_n^l(v)), \tag{7a}$$

$$f_n^{l+1}(v) = F_{\text{updating}}^l\left(\{m_n^l(u, v); u \in \mathcal{N}(v)\}\right), \tag{7b}$$

where $F_{\text{message}}^l$ and $F_{\text{updating}}^l$ are the message function and updating function in the $l$-th round, and $\mathcal{N}(v)$ is the neighborhood of the node $v$. In our implementations, the choices of $F_{\text{message}}^l$ and $F_{\text{updating}}^l$ follow the edge convolution operator (Wang et al., 2018), while other variants of GNNs can also be applied. The final node features of $\tau_n$ are given by $\{f_n^L(v) : v \in V_n\}$ after $L$ rounds of message passing.

**Node hidden states**  The conditional distribution $Q(\cdot|e_{<n})$ is highly complicated as it has to capture how $x_{n+1}$ can be added to $\tau_n$ based on how previous leaf nodes are added to form the tree topologies. A common approach is to use RNNs to model this complex distribution that strikes a good balance between expressiveness and scalability (You et al., 2018b; Liao et al., 2019). In our model, after obtaining the final node features of $\tau_n$, a gated recurrent unit (GRU) (Cho et al., 2014) follows, i.e.

$$h_n(v) = \text{GRU}(h_{n-1}(v), f_n^L(v)), \tag{8}$$

where $h_n(v)$ is the hidden state of $v$ at the $n$-th generation step and is initialized to zero for the newly added nodes including those in $\tau_3$. The node hidden states $\{h_n(v); v \in V_n\}$, therefore, contain the information of all the tree topologies generated so far which can be used for conditional distribution modeling.

**Time guided readout**  We now construct the distribution $Q(\cdot|e_{<n})$ over edge decisions based on the node hidden states. As mentioned before, a main difference between our model and other graph autoregressive models is that the node embedding $f_n^0(v)$ of a node $v$ may vary with the time step $n$. We, therefore, incorporate time embeddings into the readout step which first forms the edge features $r_n(e) \in \mathbb{R}$ of $e = (u, v)$ using

$$p_n(e) = F_{\text{pooling}}\left(h_n(u) + b_n, h_n(v) + b_n\right), \tag{9a}$$
$$r_n(e) = F_{\text{readout}}\left(p_n(e) + b_n\right), \tag{9b}$$

where $b_n$ is the sinusoidal positional embedding of time step $n$ that is widely used in Transformers (Vaswani et al., 2017), $F_{\text{pooling}}$ is the pooling function implemented as 2-layer MLPs followed by an elementwise maximum operator, and $F_{\text{readout}}$ is the readout function implemented as 2-layer MLPs with a scalar output. Then the conditional distribution for edge decision is

$$Q(\cdot|e_{<n}) \sim \text{Discrete}(q_n), \quad q_n = \text{softmax}\left(\{r_n(e)\}_{e \in E_n}\right), \tag{10}$$

where probability vector $q_n \in \mathbb{R}^{|E_n|}$ for parametrizing $Q(\cdot|e_{<n})$ is obtained by applying a softmax function to all the time guided edge features in equation (9b).

As the last step, we sample an edge $e_n \in E_n$ from the discrete distribution in equation (10) and add the leaf node $x_{n+1}$ to $e_n$ as described in Section 3.1. This way, we update the ordinal tree topology from $\tau_n$ of rank $n$ to $\tau_{n+1}$ of rank $n + 1$. We can repeat this procedure until an ordinal tree topology $\tau = \tau_N$ of rank $N$ is reached. The probability of $\tau$ then takes the form

$$Q_\phi(\tau) = Q_\phi(D) = \prod_{n=3}^{N-1} Q_\phi(e_n|e_{<n}), \tag{11}$$

where $D$ is the decision sequence and $\phi$ are the learnable parameters in the model. We call this autoregressive model for tree topologies ARTree, and summarize it in Algorithm 1. Note that equation (11) can also be used for tree topology probability evaluation where the decision sequence $D = g^{-1}(\tau)$ is obtained from the decomposition process (Appendix C) that enjoys a linear time complexity (Lemma 1). Compared to SBNs, ARTree does not rely on heuristic features for parameterization and can provide distributions whose support spans the entire tree topology space as all possible decisions would have nonzero probabilities due to the softmax parameterization in equation (10). Although different taxa orders may affect the performance of ARTree, we find this effect is negligible in our experiments.

There exist other VI approaches for phylogenetic inference that also use unconfined models over the space of tree topologies. Moretti et al. (2021) proposed to sample tree topologies through subtree merging and resampling following CSMC (Wang et al., 2015), but employed a parametrized proposal distribution. Koptagel et al. (2022) proposed a novel multifurcating tree topology sampler named SLANTIS, which makes decisions on adding edges in a specific order based on a simply parameterized weight matrix and maximum spanning trees, and is integrated with CSMC to sample bifurcating tree

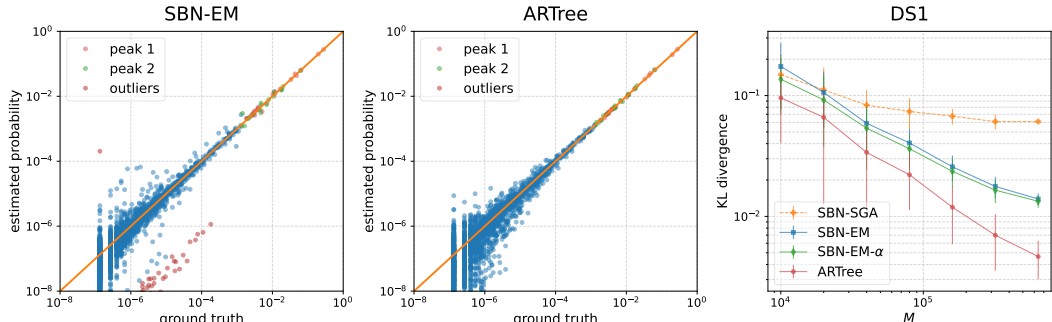

Figure 2: Performances of different methods for TDE on DS1. **Left/Middle**: Comparison of the ground truth and the estimated probabilities using SBN-EM and ARTree. A tree topology is marked as an outlier if it satisfies $|\log(\text{estimated probability}) - \log(\text{ground truth})| > 2$. **Right**: The KL divergence as a function of the sample size. The results are averaged over 10 replicates with one standard deviation as the error bar.

topologies ($\phi$-CSMC). Unlike these methods, ARTree employs GNNs for an autoregressive model that builds up the tree topology sequentially through leaf node addition operations, which not only allows fast sampling of trees, but also provides straightforward density estimation procedures. We demonstrate the advantage of ARTree over these baselines in the experiments.

## 4 Experiments

In this section, we test the effectiveness and efficiency of ARTree for phylogenetic inference on two benchmark tasks: tree topology density estimation (TDE) and variational Bayesian phylogenetic inference (VBPI). In all experiments, we report the inclusive KL divergence from posterior estimates to the ground truth to measure the approximation error of different methods. We will use "KL divergence" for inclusive KL divergence throughout this section unless otherwise specified. The code is available at `https://github.com/tyuxie/ARTree`.

**Experimental setup**  We perform experiments on eight data sets which we will call DS1-8. These data sets, consisting of sequences from 27 to 64 eukaryote species with 378 to 2520 site observations, are commonly used to benchmark phylogenetic MCMC methods (Hedges et al., 1990; Garey et al., 1996; Yang & Yoder, 2003; Henk et al., 2003; Lakner et al., 2008; Zhang & Blackwell, 2001; Yoder & Yang, 2004; Rossman et al., 2001; Höhna & Drummond, 2012; Larget, 2013; Whidden & Matsen IV, 2015). For the Bayesian setting, we focus on the joint posterior distribution of the tree topologies and the branch lengths and assume a uniform prior on the tree topologies, an i.i.d. exponential prior $\text{Exp}(10)$ on branch lengths, and the simple JC substitution model (Jukes et al., 1969). For each of these data sets, we run 10 single-chain MrBayes (Ronquist et al., 2012) for one billion iterations, collect samples every 1000 iterations, and discard the first $25\%$ samples as burn-in. These samples form the ground truth of the marginal distribution of tree topologies to which we will compare the posterior estimates obtained by different methods. All GNNs have $L = 2$ rounds in the message passing step. All the activation functions in MLPs are exponential linear units (ELUs) (Clevert et al., 2015). The taxa order is set to the lexicographical order of the corresponding species names in all experiments except the ablation studies. All the experiments are run on an Intel Xeon Platinum 9242 processor. All models are implemented in PyTorch (Paszke et al., 2019) and trained with the Adam (Kingma & Ba, 2015) optimizer. The learning rate is 0.001 for SBNs, 0.0001 for ARTree, and 0.001 for the branch length model.

### 4.1 Tree topology density estimation

We first investigate the performance of ARTree for tree topology density estimation given the MCMC posterior samples on DS1-8. Following Zhang & Matsen IV (2018), we run MrBayes on each data set with 10 replicates of 4 chains and 8 runs until the runs have ASDSF (the standard convergence criteria used in MrBayes) less than 0.01 or a maximum of 100 million iterations. The training data

Table 1: KL divergences to the ground truth of different methods across 8 benchmark data sets. Sampled trees column shows the numbers of unique tree topologies in the training sets formed by MrBayes runs. The results are averaged over 10 replicates. The results of SBN-EM, SBN-EM-$\alpha$ are from Zhang & Matsen IV (2018).

| Data set | #Taxa | #Sites | Sampled trees | KL divergence to ground truth | | | |
|---|---|---|---|---|---|---|---|
| | | | | SBN-EM | SBN-EM-$\alpha$ | SBN-SGA | ARTree |
| DS1 | 27 | 1949 | 1228 | 0.0136 | 0.0130 | 0.0504 | **0.0045** |
| DS2 | 29 | 2520 | 7 | 0.0199 | 0.0128 | 0.0118 | **0.0097** |
| DS3 | 36 | 1812 | 43 | 0.1243 | 0.0882 | 0.0922 | **0.0548** |
| DS4 | 41 | 1137 | 828 | 0.0763 | 0.0637 | 0.0739 | **0.0299** |
| DS5 | 50 | 378 | 33752 | 0.8599 | 0.8218 | 0.8044 | **0.6266** |
| DS6 | 50 | 1133 | 35407 | 0.3016 | 0.2786 | 0.2674 | **0.2360** |
| DS7 | 59 | 1824 | 1125 | 0.0483 | 0.0399 | 0.0301 | **0.0191** |
| DS8 | 64 | 1008 | 3067 | 0.1415 | 0.1236 | 0.1177 | **0.0741** |

sets are formed by collecting samples every 100 iterations and discarding the first 25%. Now, given a training data set $\mathcal{M} = \{\tau_m\}_{m=1}^{M}$, we train ARTree via maximum likelihood estimation using stochastic gradient ascent. In each iteration, the stochastic gradient is obtained as follows

$$\nabla_{\phi} L(\phi; \mathcal{M}) = \frac{1}{B} \sum_{b=1}^{B} \nabla_{\phi} \log Q_{\phi}(\tau_{m_b}),$$ (12)

where a minibatch $\{\tau_{m_b}\}_{b=1}^{B}$ is randomly sampled from $\mathcal{M}$. We compare ARTree to SBN baselines including SBN-EM, SBN-EM-$\alpha$, and SBN-SGA. For SBN-EM and SBN-EM-$\alpha$, we use the same setting as previously done in Zhang & Matsen IV (2018) (see Appendix A for more details). In addition to these EM variants, a gradient based method for SBNs called SBN-SGA is considered, where SBNs are reparametrized with the latent parameters initialized as zero (see equation (18) in Appendix B) and optimized via stochastic gradient ascent, similarly to ARTree. For both ARTree and SBN-SGA, the results are collected after 200000 parameter updates with batch size $B = 10$.

The left and middle plots of Figure 2 show a comparison between ARTree and SBN-EM on DS1, which has a peaky posterior distribution. Compared to SBN-EM, ARTree provides more accurate probability estimates for tree topologies on the peaks and significantly reduces the large biases in the low probability region (the crimson dots). The right plot of Figure 2 shows the KL divergence of different methods as a function of the sample size of the training data. We see that ARTree consistently outperforms SBN based methods for all $M$s. Moreover, as the sample size $M$ increases, ARTree keeps providing better approximation while SBNs start to level off when $M$ is large. This indicates the superior flexibility of ARTree over SBNs for tree topology density estimation.

Table 1 shows the KL divergences of different methods on DS1-8. We see that ARTree outperforms SBN based methods on all data sets. The gradient based method SBN-SGA is better than SBN-EM on most of the data sets because SBN-EM is well initialized (Zhang & Matsen IV, 2018) and more likely to get trapped in local modes. From this point of view, the comparison between ARTree and SBN-SGA is fair because they both use a uniform initialization that facilitates exploration.

### 4.2 Variational Bayesian phylogenetic inference

Our second experiment is on VBPI, where we compare ARTree to SBNs for tree topology variational approximations. Both methods are evaluated on the aforementioned benchmark data sets DS1-8. Following Zhang & Matsen IV (2019), we use the simplest SBN and gather the subsplit support from 10 replicates of 10000 ultrafast maximum likelihood bootstrap trees (Minh et al., 2013). For both ARTree and SBNs, the collaborative branch lengths are parametrized using the learnable topological features with the edge convolution operator (EDGE) for GNNs (Zhang, 2023). We set $K = 10$ for the multi-sample lower bound (4) and use the following annealed unnormalized posterior at the $i$-th iteration

$$p(\boldsymbol{Y}, \tau, \boldsymbol{q}; \beta_i) = p(\boldsymbol{Y}|\tau, \boldsymbol{q})^{\beta_i} p(\tau, \boldsymbol{q})$$ (13)

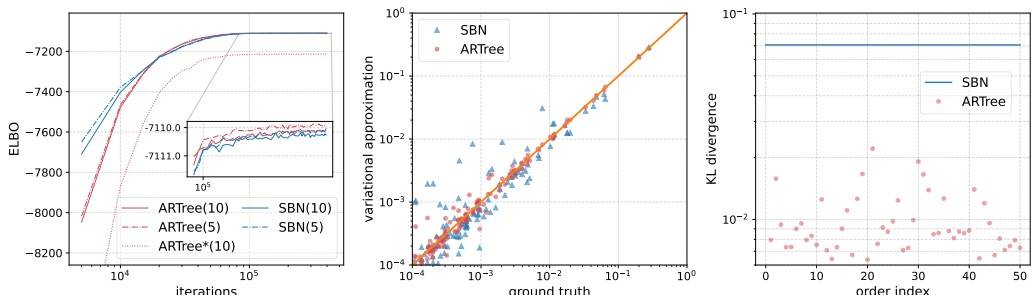

Figure 3: Performances of ARTree and SBN as tree topology variational approximations for VBPI on DS1. **Left**: the evidence lower bound (ELBO) as a function of iterations. The numbers of particles used in the training objective are in the brackets. The ARTree* method refers to ARTree without time guidance, i.e. $b_n = 0$ for all $n$ in the readout step. **Middle**: variational approximations vs ground truth posterior probabilities of the tree topologies. **Right**: KL divergences across 50 random taxa orders. The KL divergence of SBNs is averaged over 10 independent trainings.

Table 2: KL divergences to the ground truth, evidence lower bound (ELBO), 10-sample lower bound (LB-10), and marginal likelihood (ML) estimates of different methods across 8 benchmark data sets. GT trees row shows the number of unique tree topologies in the ground truth. The marginal likelihood estimates are obtained via importance sampling using 1000 samples. The KL results are averaged over 10 independent trainings. For ELBO, LB-10, and ML, the results are averaged over 100, 100, and 1000 independent runs respectively with standard deviation in the brackets. The results of $\phi$-CSMC are from Koptagel et al. (2022). For ELBO and LB-10, a larger mean is better; for ML, a smaller standard deviation is better[1].

| | Data set | DS1 | DS2 | DS3 | DS4 | DS5 | DS6 | DS7 | DS8 |
|---|---|---|---|---|---|---|---|---|---|
| | # Taxa | 27 | 29 | 36 | 41 | 50 | 50 | 59 | 64 |
| | # Sites | 1949 | 2520 | 1812 | 1137 | 378 | 1133 | 1824 | 1008 |
| | GT trees | 2784 | 42 | 351 | 11505 | 1516877 | 809765 | 11525 | 82162 |
| KL | SBN | 0.0707 | 0.0144 | 0.0554 | 0.0739 | 1.2472 | 0.3795 | 0.1531 | 0.3173 |
| | ARTree | **0.0097** | **0.0004** | **0.0064** | **0.0219** | **0.8979** | **0.2216** | **0.0123** | **0.1231** |
| ELBO | SBN | -7110.24(0.03) | -26368.88(0.03) | -33736.22(0.02) | -13331.83(0.03) | -8217.80(0.04) | **-6728.65(0.06)** | -37334.85(0.04) | -8655.05(0.05) |
| | ARTree | **-7110.09(0.04)** | **-26368.78(0.07)** | **-33736.17(0.08)** | **-13331.82(0.05)** | **-8217.68(0.04)** | **-6728.65(0.06)** | **-37334.84(0.13)** | **-8655.03(0.05)** |
| LB-10 | SBN | -7108.69(0.02) | -26367.87(0.02) | -33735.26(0.02) | -13330.29(0.02) | -8215.42(0.04) | -6725.33(0.04) | -37332.58(0.03) | -8651.78(0.04) |
| | ARTree | **-7108.68(0.02)** | **-26367.86(0.02)** | **-33735.25(0.02)** | **-13330.27(0.03)** | **-8215.34(0.03)** | **-6725.33(0.04)** | **-37332.54(0.03)** | **-8651.73(0.04)** |
| ML | $\phi$-CSMC | -7290.36(7.23) | -30568.49(31.34) | -33798.06(6.62) | -13582.24(35.08) | -8367.51(8.87) | -7013.83(16.99) | N/A | -9209.18(18.03) |
| | SBN | **-7108.41(0.15)** | -26367.71(0.08) | **-33735.09(0.09)** | -13329.94(0.20) | -8214.62(0.40) | **-6724.37(0.43)** | -37331.97(0.28) | -8650.64(0.50) |
| | ARTree | **-7108.41(0.19)** | **-26367.71(0.07)** | **-33735.09(0.09)** | **-13329.94(0.17)** | **-8214.59(0.34)** | -6724.37(0.46) | **-37331.95(0.27)** | **-8650.61(0.48)** |

where $\beta_i = \min\{1.0, 0.001 + i/H\}$ is the inverse temperature that goes from 0.001 to 1 after $H$ iterations. For ARTree, a long annealing period $H = 200000$ is used for DS6 and DS7 due to the highly multimodal posterior distributions on these two data sets (Whidden & Matsen IV, 2015) and $H = 100000$ is used for the other data sets. For SBNs, we set $H = 100000$ for all data sets. The Monte Carlo gradient estimates for the tree topology parameters and the branch lengths parameters are obtained via VIMCO (Mnih & Rezende, 2016) and the reparametrization trick (Zhang & Matsen IV, 2019) respectively. The results are collected after 400000 parameter updates.

The left plot in Figure 3 shows the evidence lower bound (ELBO) as a function of the number of iterations on DS1. Although the larger support of ARTree adds to the complexity of training for tree topology variational approximation, we see that by the time SBN based methods converge, ARTree based methods achieve comparable (if not better) lower bounds and finally surpass the SBN baselines in the end. We also find that using fewer particles $(K = 5)$ in the training objective tends

---

[1] We use standard deviation as a criterion because in the context of importance sampling, the variance of an estimator reflects the approximation accuracy of the importance distribution to the target. Note that this is only reasonable if the importance distribution is proper (so that the marginal likelihood is unbiased), and hence does not apply to $\phi$-CSMC even though it also has much larger variance.

to provide larger ELBO. Moreover, time guidance turns out to be crucial for ARTree, as evidenced by the significant performance drop when it is turned off. As shown in the middle plot, compared to SBNs, ARTree can provide a more accurate variational approximation of the tree topology posterior. To investigate the effect of taxa orders on ARTree, we randomly sample 50 taxa orders and report the KL divergence for each order in the right plot of Figure 3. We find that ARTree exhibits weak randomness as the taxa order varies and consistently outperforms SBNs by a large margin.

Table 2 shows the KL divergences to the ground truth, evidence lower bound (ELBO), 10-sample lower bound (LB-10), and marginal likelihood (ML) estimates obtained by different methods on DS1-8. We find that ARTree achieves smaller KL divergences than SBNs across all data sets and performs on par or better than SBNs for lower bound and marginal likelihood estimation. Compared to SBNs, the ELBOs provided by ARTree tend to have larger variances, especially on DS2, DS3, and DS7, which is partly due to the larger support of ARTree that spans the entire tree topology space (see more discussions in Appendix E).

## 5   Conclusion

In this paper, we introduced ARTree, a deep autoregressive model over tree topologies for phylogenetic inference. Unlike SBNs that rely on hand-engineered features for parameterization and require pre-sampled tree topologies, ARTree is built solely on top of learnable topological features (Zhang, 2023) via GNNs which allows for a rich family of distributions over the entire phylogenetic tree topology space. Moreover, as an autoregressive model, ARTree also allows simple forward sampling procedures and straightforward density computation, which make it readily usable for tree topology density estimation and variational Bayesian phylogenetic inference. In experiments, we showed that ARTree outperforms SBNs on a benchmark of challenging real data tree topology density estimation and variational Bayesian phylogenetic inference problems, especially in terms of tree topology posterior approximation accuracy.

## Acknowledgements

This work was supported by National Natural Science Foundation of China (grant no. 12201014 and grant no. 12292983), as well as National Institutes of Health grant AI162611. The research of Cheng Zhang was support in part by National Engineering Laboratory for Big Data Analysis and Applications, the Key Laboratory of Mathematics and Its Applications (LMAM) and the Key Laboratory of Mathematical Economics and Quantitative Finance (LMEQF) of Peking University. The authors are grateful for the computational resources provided by the High-performance Computing Platform of Peking University. The authors appreciate the anonymous NeurIPS reviewers for their constructive feedback.

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

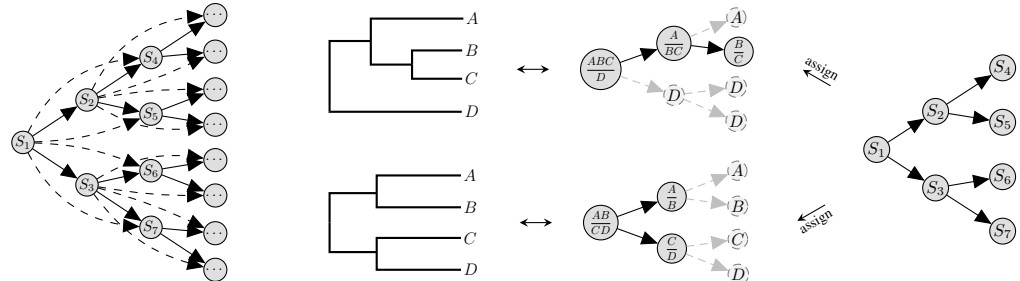

Figure 4: Subsplit Bayesian networks and a simple example for a leaf set of 4 taxa (denoted by $A, B, C, D$ respectively). **Left:** General subsplit Bayesian networks. The solid full and complete binary tree network is $B_{\mathcal{X}}^*$. The dashed arrows represent the additional dependence for more expressiveness. **Middle Left:** Examples of (rooted) phylogenetic trees that are hypothesized to model the evolutionary history of the taxa. **Middle Right:** The corresponding subsplit assignments for the trees. For ease of illustration, subsplit $(Y, Z)$ is represented as $\frac{Y}{Z}$ in the graph. **Right:** The SBN for this example, which is $\mathcal{B}_{\mathcal{X}}^*$ in this case.

## A Details of subsplit Bayesian networks

One recent and expressive graphical model that provides a flexible family of distributions over tree topologies is the subsplit Bayesian network, as proposed by Zhang & Matsen IV (2018). Let $\mathcal{X}$ be the set of $N$ labeled leaf nodes. A non-empty set $C$ of $\mathcal{X}$ is referred to as a *clade* and the set of all clades of $\mathcal{X}$, denoted by $\mathcal{C}(\mathcal{X})$, is a totally ordered set with a partial order $\succ$ (e.g., lexicographical order) defined on it. An ordered pair of clades $(W, Z)$ is called a *subsplit* of a clade $C$ if it is a bipartition of $C$, i.e., $W \succ Z$, $W \cap Z = \emptyset$, and $W \cup Z = C$.

**Definition 2** (Subsplit Bayesian Network). *A subsplit Bayesian network (SBN) $\mathcal{B}_{\mathcal{X}}$ on a leaf node set $\mathcal{X}$ of size $N$ is defined as a Bayesian network whose nodes take on subsplit or singleton clade values of $\mathcal{X}$ and has the following properties: (a) The root node of $\mathcal{B}_{\mathcal{X}}$ takes on subsplits of the entire labeled leaf node set $\mathcal{X}$; (b) $\mathcal{B}_{\mathcal{X}}$ contains a full and complete binary tree network $B_{\mathcal{X}}^*$ as a subnetwork; (c) The depth of $B_{\mathcal{X}}$ is $N - 1$, with the root counted as depth 1.*

Due to the binary structure of $B_{\mathcal{X}}^*$, the nodes in SBNs can be indexed by denoting the root node with $S_1$ and two children of $S_i$ with $S_{2i}$ and $S_{2i+1}$ recursively where $S_i$ is an internal node (see the left plot in Figure 4). For any rooted tree topology, by assigning the corresponding subsplits or singleton clades values $\{S_i = s_i\}_{i \geq 1}$ to its nodes, one can uniquely map it into an SBN node assignment (see the middle and right plots in Figure 4).

As Bayesian networks, the SBN based probability of a rooted tree topology $\tau$ takes the following form

$$p_{\mathrm{sbn}}(T = \tau) = p(S_1 = s_1) \prod_{i > 1} p(S_i = s_i | S_{\pi_i} = s_{\pi_i}), \tag{14}$$

where $\pi_i$ is the index set of the parents of node $i$. For unrooted tree topologies, we can also define their SBN based probabilities by viewing them as rooted tree topologies with unobserved roots and integrating out the positions of the root node as follows:

$$p_{\mathrm{sbn}}(T^{\mathrm{u}} = \tau) = \sum_{e \in E(\tau)} p_{\mathrm{sbn}}(\tau^e) \tag{15}$$

where $\tau^e$ is the resulting rooted tree topology when the rooting position is on edge $e$.

In practice, SBNs are parameterized according to the *conditional probability sharing* principle where the conditional probability for parent-child subsplit pairs are shared across the SBN network, regardless of their locations. The set of all conditional probabilities are called conditional probability tables (CPTs). Parameterizing SBNs, therefore, often requires finding an appropriate support of CPTs. For tree topology density estimation, this can be done using the sample of tree topologies that is given as the data set. For variational Bayesian phylogenetic inference, as no sample of tree topologies is available, one often resorts to fast bootstrap or MCMC methods (Minh et al., 2013; Zhang, 2020). Let $\mathbb{S}_{\mathrm{r}}$ denotes the root subsplits and $\mathbb{S}_{\mathrm{ch|pa}}$ denotes the child-parent subsplit pairs in the support. The parameters of SBNs are then $p = \{p_{s_1}; s_1 \in \mathbb{S}_{\mathrm{r}}\} \cup \{p_{s|t}; s|t \in \mathbb{S}_{\mathrm{ch|pa}}\}$ where

$$p_{s_1} = p(S_1 = s_1), \quad p_{s|t} = p(S_i = s | S_{\pi_i} = t), \ \forall i > 1. \tag{16}$$

As a result, the supports of SBN-induced distributions are often limited by the splitting patterns in the observed samples and could not span the entire tree topology space (Zhang & Matsen IV, 2022).

**The SBN-EM Algorithm**   For unrooted tree topologies, the SBN based probability (15) can be viewed as a hidden variable model where the root subsplit is the hidden variable. In this case, SBNs can be trained using the expectation-maximization (EM) algorithm, as proposed by Zhang & Matsen IV (2018). Given a training set $\{\tau_k\}_{k=1}^M$, we first initialize the parameter estimates as $\hat{p}^{EM,(0)}$ (i.e., the simple average estimates as in Zhang & Matsen IV (2018)). In the $i$-th step, we run the E-step and M-step as follows

- **E-step**: $\forall 1 \leq k \leq M$, compute $q_k^{(i)}(s_1) = \frac{p(\tau_k, s_1 | \hat{p}^{EM,(i)})}{\sum_{s_1 \sim \tau_k} p(\tau_k, s_1 | \hat{p}^{EM,(i)})}$ where $s_1 \sim \tau_k$ means the subsplit $s_1$ can be achieved by placing a "virtual root" on an edge of $\tau$.
- **M-step**: update the parameter estimates by

$$\hat{p}_{s_1}^{EM,(i+1)} = \frac{\bar{m}_{s_1}^{(i)} + \alpha \tilde{m}_{s_1}}{K + \alpha \sum_{s_1 \in \mathbb{S}_r} \tilde{m}_{s_1}}, \quad \bar{m}_{s_1}^{(i)} = \sum_{k=1}^M \sum_{e \in E(\tau_k)} q_k^{(i)}(s_1) \, \mathbb{I}\left(s_{1,k}^e = s_1\right)$$

$$\hat{p}_{s|t}^{EM,(i+1)} = \frac{\bar{m}_{s,t}^{(i)} + \alpha \tilde{m}_{s,t}}{\sum_s \left(\bar{m}_{s,t}^{(i)} + \alpha \tilde{m}_{s,t}\right)}, \quad \bar{m}_{s,t}^{(i)} = \sum_{k=1}^M \sum_{e \in E(\tau_k)} q_k^{(i)}\left(s_{1,k}^e\right) \sum_{j>1} \mathbb{I}\left(s_{j,k}^e = s, s_{\pi_j,k}^e = t\right)$$

where $\mathbb{I}$ is the indicator function, $s_{j,k}^e$ is the node value of $S_j$ in $\tau_k^e$, $\tilde{m}_s$ and $\tilde{m}_{s,t}$ are equivalent counts and $\alpha$ is the regularization coefficient that encourages generalization.

When $\alpha > 0$, this algorithm is called SBN-EM-$\alpha$.

# B   Details of variational Bayesian phylogenetic inference

With two variational families $Q_\phi(\tau)$ and $Q_\psi(q|\tau)$ over the space of tree topologies and branch lengths, the variational Bayesian phylogenetic inference (VBPI) approach forces $Q_{\phi,\psi}(\tau, q) = Q_\phi(\tau)Q_\psi(q|\tau)$ to approximate the posterior $p(\tau, q|Y)$ by maximizing the following multi-sample lower bound

$$L^K(\phi, \psi) = \mathbb{E}_{\{(\tau^i, q^i)\}_{i=1}^K \overset{\text{i.i.d.}}{\sim} Q_{\phi,\psi}} \log \left( \frac{1}{K} \sum_{i=1}^K \frac{p(Y|\tau^i, q^i)p(\tau^i, q^i)}{Q_\phi(\tau^i)Q_\psi(q^i|\tau^i)} \right). \tag{17}$$

Gradients of the objective (17) w.r.t. $\phi$ and $\psi$ can be estimated by the VIMCO estimator (Mnih & Rezende, 2016) and the reparameterization trick respectively. In the following, we introduce some common choices of $Q_\phi(\tau)$ and $Q_\psi(q|\tau)$.

**Choice of $Q_\phi(\tau)$**   Before the proposed ARTree framework in this article, SBNs is the common choice of $Q_\phi(\tau)$. As introduced in Appendix A, SBNs provide a probability distribution over unrooted tree topologies in equation (15). Given a subsplit support of CPTs, SBNs can be parameterized as follows

$$p_{s_1} = \frac{\exp(\phi_{s_1})}{\sum_{s' \in \mathbb{S}_r} \exp(\phi_{s'})}, \ s_1 \in \mathbb{S}_r; \quad p_{s|t} = \frac{\exp(\phi_{s|t})}{\sum_{s':s'|t \in \mathbb{S}_{ch|pa}} \exp(\phi_{s'|t})}, \ s|t \in \mathbb{S}_{ch|pa}. \tag{18}$$

The parameters $\phi = \{\phi_{s_1}; \ s_1 \in \mathbb{S}_r\} \cup \{\phi_{s|t}; \ s|t \in \mathbb{S}_{ch|pa}\}$ are called latent parameters of SBNs.

**Choice of $Q_\psi(q|\tau)$**   The distribution $Q_\psi(q|\tau)$ is often taken to be a diagonal lognormal distribution, which can be parametrized using some heuristic features (Zhang & Matsen IV, 2019) or the recently proposed learnable topological features (Zhang, 2023) of $\tau$ as follows. For each edge $e = (u, v)$ in $\tau$, one can first obtain the edge features using $h_e = f(h_u, h_v)$ where $h_u$ is the GNN output at node $u$ and $f$ is a permutation invariant function. Then the mean and standard deviation parameters are given by

$$\mu(e, \tau) = \text{MLP}^\mu(h_e), \quad \sigma(e, \tau) = \text{MLP}^\sigma(h_e)$$

where $\text{MLP}^\mu$ and $\text{MLP}^\sigma$ are two multi-layer perceptrons (MLPs).

---
**Algorithm 2:** Tree topology decomposition process

---
    **Input:** a tree topology $\tau$ with all of the $N$ leaf nodes.
    **Output:** a decision sequence $D$.
    $\tau_N = (V_N, E_N) \leftarrow$ the tree topology $\tau$;
    **for** $n = N - 1, \ldots, 3$ **do**
       |   Determine the unique neighbor $w$ of the leaf node $x_{n+1}$;
       |   Determine the two neighbors $u$ and $v$ (except $x_{n+1}$) of $w$;
       |   $V_n \leftarrow V_{n+1} \backslash \{w, x_{n+1}\}$;
       |   $E_n \leftarrow (E_{n+1} \cup \{(u,v)\}) \backslash \{(w, x_{n+1}), (w, u), (w, v)\}$;
       |   $\tau_n \leftarrow (V_n, E_n)$;
       |   $e_n \leftarrow (u, v)$;
    **end**
    $D \leftarrow (e_3, \ldots, e_{N-1})$.

---

## C  Details of tree topology decomposition process

The tree topology decomposition process, which maps a tree topology to a corresponding decision sequence, is indeed the inverse operation of Algorithm 1. Also, the decomposition process is implemented in a recursive way starting from the tree topology $\tau_N$ of rank $N$. Intuitively, given an ordinal tree topology $\tau_{n+1}$ of rank $n + 1$, one can detach the leaf node $x_{n+1}$ as well as its unique neighbor $w$ and reconnect the two neighbors of $w$. The remaining graph, denoted by $\tau_n$, is an ordinal tree topology of rank $n$; the edge decision $e_n$ is exactly the reconnected edge. This process continues until the unique ordinal tree topology $\tau_3$ of rank 3 is reached. We summarize the sketch of tree topology decomposition process in Algorithm 2.

Given a tree topology $\tau$ with all of the $N$ leaf nodes, we can evaluate its ARTree based probability by first mapping it to a decision sequence $D = (e_3, \ldots, e_{N-1})$ following Algorithm 2 and then calculate the probability as the product of conditionals

$$Q(\tau) = Q(D) = \prod_{n=3}^{N-1} Q(e_n | e_{<n}).$$

## D  The proofs of Theorem 1 and Lemma 1

**Theorem 1.** *Let $\mathcal{D} = \{D | D = (e_3, \ldots, e_{N-1}), \ e_n \in E_n, \forall \ 3 \leq n \leq N - 1\}$ be the set of all decision sequences of length $N - 3$ and $\mathcal{T}$ be the set of all ordinal tree topologies of rank $N$. Let the map*

$$g: \quad \mathcal{D} \quad \to \quad \mathcal{T}$$
$$D \quad \mapsto \quad \tau$$

*be the generating process used in ARTree. Then $g$ is a bijection between $\mathcal{D}$ and $\mathcal{T}$.*

**Proof of Theorem 1** It is obvious that $g$ is a well-defined map from $\mathcal{D}$ to $\mathcal{T}$. To prove it is a bijection, it suffices to show $g$ is injective and surjective.

We first show that $g$ is injective. Assume there are two decision sequences $D^{(1)}, D^{(2)} \in \mathcal{D}$ and $D^{(1)} \neq D^{(2)}$. Let $k$ be the first position where $D^{(1)}$ and $D^{(2)}$ begin to differ, i.e. $e_i^{(1)} = e_i^{(2)}$ for all $i < k$ and $e_k^{(1)} \neq e_k^{(2)}$. If $g(D^{(1)}) = g(D^{(2)}) =: \tau$, one can take the subtree topology of rank $k$ and $k + 1$ of $\tau$, $\tau_k$ and $\tau_{k+1}$. Noting that $e_k$ refers to the edge in $\tau$ where to add the new node $x_{n+1}$, the equation $e_k^{(1)} \neq e_k^{(2)}$ implies they will induce different $\tau_{k+1}$s. This contradicts the uniqueness of $\tau_{k+1}$. Therefore, we conclude that $g$ is injective.

Next, we prove that $g$ is surjective. For a tree topology $\tau$ with rank $N$, we denote its subtree topology of rank $k$ by $\tau_k$, where $k = 3, \ldots, n$. In fact, for each $k$, the tree topology $\tau_{k+1}$ corresponds to adding the leaf node $x_{k+1}$ to an edge in $\tau_k$, and we denote this edge by $e_k$. It is easy to verify that the constructed $D = (e_3, \ldots, e_{N-1})$ is a preimage of $\tau$.

Table 3: The ELBO estimates on DS1 obtained by different combinations of tree topology model $Q(\tau)$ and branch length model $Q(\boldsymbol{q}|\tau)$. The results are averaged over 100 independent runs with standard deviation in the brackets.

| Model combination | ELBO |
|---|---|
| SBN + branch length model trained along with SBN | -7110.24(0.03) |
| SBN + branch length model trained along with ARTree | -7110.26(0.03) |
| ARTree + branch length model trained along with ARTree | -7110.09(0.04) |

Table 4: Runtime comparison in the variational inference setting on DS1[2]. SBN* and ARTree* refer to the early stopping of SBN and ARTree that surpass the CSMC baseline in terms of marginal likelihood estimation (-7290.36), respectively. The experiments are run on a single core of MacBook Pro 2019.

| | VCSMC | VaiPhy | $\phi$-CSMC | SBN | ARTree | SBN* | ARTree* |
|---|---|---|---|---|---|---|---|
| Total training time (minutes) | 248.3 | 45.1 | N/A | 659.3 | 3740.8 | 10.2 | 79.5 |
| Evaluation time (one estimate of ML, minutes) | 2.4 | 1.6 | 102.2 | 0.15 | 0.41 | 0.15 | 0.41 |

**Lemma 1.** *The time complexity of the decomposition process induced by $g^{-1}(\cdot)$ is $O(N)$.*

**Proof of Lemma 1**    Assume the tree topology $\tau$ is stored as a binary tree data structure, where each node other than the root node also has a parent node pointer. Before decomposing $\tau$, we first build a dictionary of the $(n, x_n)$ mappings by a traversal across $\tau$ that maps $n$ to the leaf node $x_n$ in $\tau$, $\forall n \leq N$. The time complexity of building this dictionary is $O(N)$. In the decomposition process, given an ordinal tree topology of rank $n+1$, it costs $O(1)$ time to locate $x_{n+1}$ and determine the unique parent node $w$ of $x_{n+1}$ and the neighbors of $w$. It is obvious that the time complexity of detaching and reconnecting operations is $O(1)$. One can repeat this procedure for $n = N-1, \ldots, 3$, resulting in a time complexity of $O(N)$. Therefore, the time complexity of the decomposition process induced by $g^{-1}(\cdot)$ is $O(N)$.

# E    Limitations

**Larger variances of ELBO estimates**    As an autoregressive model for phylogenetic tree topology, ARTree provides reliable approximations for target distributions over tree topologies, as evidenced by our real data experiments. However, when incorporated with the branch length model for VBPI, the ELBOs provided by ARTree tend to have larger variances, which we find is caused by the occasional occurrence of "outliers" among samples. In fact, as the support of ARTree spans over the entire tree topology space, it adds to the difficulty of fitting the conditional distribution $Q_\psi(\boldsymbol{q}|\tau)$, compared to SBNs. When combined with ARTree, the approximation accuracy of $Q_\psi(\boldsymbol{q}|\tau)$ might be related to the cluster structure of peaks in the tree topology posterior. See Figure 3 in Whidden & Matsen IV (2015) for cluster subtree-prune-and-regraft (SPR) graphs of DS1-8. We also examine the approximation accuracies of $Q_\psi(\boldsymbol{q}|\tau)$ trained along with ARTree for those $\tau$ in the support of SBNs and find a significant enhancement in ELBO and reduction of variances. This phenomenon raises an important topic: proper design and optimization of branch length model when the support of tree topology model spans the entire space. We leave this for future work.

---

[2]**Training**: We trained all models following the setting in their original papers: VCSMC was trained for 100 iterations with 2048 particles per iteration; VaiPhy was trained for 200 iterations with 128 particles per iteration; $\phi$-CSMC directly estimates ML based on VaiPhy, and therefore does not need extra training; both ARTree and SBN were trained for 400000 iterations with 10 particles per iteration.
**Evaluation**: We find that the evaluation strategies in their original papers are quite different, e.g. VaiPhy, SBN, and ARTree used importance sampling to estimate ML with different repetition times; VCSMC and $\phi$-CSMC instead estimated ML with sequential Monte Carlo (SMC), also with different repetition times. To be fair, we report the time for producing one estimate of ML from each of these models (VaiPhy, SBN, and ARTree used importance sampling with 1000 particles; VCSMC and $\phi$-CSMC used SMC with 2048 particles).

**Minor increasement of lower bounds**  From Table 2, the estimates of lower bounds (as well as marginal likelihoods) of ARTree do not improve much over those of SBNs. We provide two explanations for this: (i) The lower bounds in VBPI are more sensitive to the quality of branch length model $Q(\boldsymbol{q}|\tau)$ instead of the tree topology model $Q(\tau)$. As SBN and ARTree use the same parametrization of the branch length model, we do not expect a large improvement in lower bounds. (ii) The support of ARTree spans the entire tree topology space. This adds to the difficulty of training $Q(\boldsymbol{q}|\tau)$ which is conditioned on tree topology $\tau$, as discussed in the above paragraph. Finally, from Table 3, we see that the enhancement on ELBOs of ARTree indeed comes from a better tree topology model as evidenced by the result of the 'SBN + branch length model trained along with ARTree' combination.

**Time complexity**  Another limitation of ARTree is its time complexity. We compare the training time and evaluation time in Table 4. Both ARTree and SBNs take more time in training than other methods with unconfined support since they both build machine-learning models with enormous parameters and rely heavily on optimization. However, they take a comparable amount of time to provide good enough approximations for marginal likelihood estimation of similar accuracy, as evidenced by ARTree$^*$ and SBN$^*$. ARTree takes more time than SBNs because it relies on several submodules which, although complicated, are designed to promote the expressive power to accommodate the complex tree space and are widely-used strategies in the literature. The inefficiency of autoregressive generative models is also an inherent issue.

