# OpenReview forum: "ARTree: A Deep Autoregressive Model for Phylogenetic Inference"
_NeurIPS.cc/2023/Conference — NeurIPS 2023 spotlight_

### Official Review · Reviewer_tL42 · 2023-07-06

**Soundness:** 4 excellent
**Presentation:** 4 excellent
**Contribution:** 4 excellent
**Rating:** 7
**Confidence:** 4

**Summary:**

This work proposes a new approach, ARTree, for obtaining more complex tree-topology approximations by combining deep autoregressive models and GNNs. The existing works in black-box VI for phylogenetic inference predominantly rely on SBNs as approximations of the tree-topology distribution, but here ARTree is shown to be superior in terms of KL divergence to the true distribution (obtained via MCMC). ARTree, in contrast to SBNs, does not rely on presampled tree topologies and thus explores the full tree-topology space, not a subset.

**Strengths:**

Due to the combinatorial nature of the tree-topology space, designing efficient and powerful density approximation algorithms of the tree-topology posterior is arguably the most complicated aspect of Bayesian phylogenetic inference. In the VI setting, SBNs have been the SOTA algorithms, but, as is mentioned in the paper, SBNs rely on presampled tree topologies. This means SBNs require candidate trees from other tree-sampling algorithms, making them not stand alone.

The algorithm proposed in the paper is an important contribution to field that is receiving increased interest in the ML community (most VI for phylogenetics papers have been published in NeurIPS, ICLR and UAI). Additionally, there have not been many attempts to improve over SBNs. As such, the proposed work appears to be relevant and could be of significant interest to the NeurIPS community.

The writing and clarity of the paper is good, although I have some questions below.

The experiments are performed using appropriate baselines (SBNs and VBPI w. SBNs). However I am missing some important references like VaiPhy (Koptagel et al., selected as oral at NeurIPS 22) and VCSMC (Moretti et al., from UAI 21). More on this below.

**Weaknesses:**

**Related work**: As this is a paper proposes a new algorithm for doing VI for phylogenetics, VaiPhy by Koptagel et al. (2022) should be referenced, preferably also VCSMC by Moretti et al. (2021). These work are highly related. I do not consider it necessary to experimentally compare ARTree with these algorithms though, as they do not compare favorably with VBPI empirically.

Furthermore, in the VaiPhy paper a sequential algorithm (SLANTIS) is designed for sampling tree topologies. I believe it can be seen as a conditional sampler in the sense that it decided whether to replace (labeled) edges in a topology based on a Bernoulli probability, given the existing edges in the topology. It is does not use GNNs, and the algorithms are clearly distinct as SLANTIS uses precomputed maximum spanning trees. However, I think a conceptual distinction should be included in the paper, nonetheless.

**Evaluating tree topologies**: It is clear that the likelihood of a topology, $\tau$, simulated by ARTree can be efficiently evaluated. At each decision, the probability of making that decision can be computed and then the probability of proposing $\tau$ is the product of these intermediate probabilities (as in Eq. 5; is this correct?).

Now, suppose I give you another tree topology, $\tau'$, that was not simulated by ARTree. Can ARTree compute the likelihood of $\tau'$, i.e. $Q(\tau')$? To me it is not clear how this is achieved by reading the paper. This seems to be like an important downstream task which SBNs can handle. If it is possible, I recommend emphasizing this feature, and how a practitioner would achieve it. If it is not possible, I think that this missing feature should be discussed.

Note that I do not regard this to be a crucial feature, ARTree is still an impressive algorithm. However, it would add transparency and promote future work on ARTree.

**"Domain expertise"**: It is repeated multiple times as a key flaw of SBNs is the required domain expertise. I do not see, and it is not explained, how ARTree diminishes this requirement? In fact, how do SBNs require more domain expertise? Running MrBAYES or UFBoot to get the presampled tree topologies can be done without understanding these algorithms in depth, as the softwares are very neatly provided. Especially, does not implementing and understanding ARTree require the same domain expertise by the practitioner as implementing and understanding SBNs?

This should be carefully clarified in the text. Alternatively, I think, the "domain expertise" argument should be removed as it does not add information as the paper is written at the moment. There are plenty of compelling arguments for ARTree over SBNs as is.

**Experiments**: In the caption of Table 2: "The KL results are averaged over 10 independent trainings." I was expecting to see standard deviations of these 10 KL numbers. How come they are not included?

Also in the same caption: "For ELBO, LB-10, and ML,the results are averaged over 100, 100, and 1000 independent runs". Do independent runs imply "independent trainings" here too? If not, why not use the 10 trained models used for the KL values to get uncertanties wrt the learned model parameters? Finally note that it says "100, 100 and 1000", which I figure is a typo.

**Stds of ML results**: I am aware that previous works reward low-variance estimators of the marginal log-likelihood, as is done here in Table 2. My guess is that this is an appropriate way to compare models that provide estimates of lower-bounds of the ML with models that can harshly over-estimate the ML, like the stepping-stone algorithm used with MrBAYES. However, here the comparison is between two VBPI models, using either ARTree or SBNs, which both use estimates of lower bounds. Could the authors please expand on why then the standard deviation is an appropriate measure of the success of the models?

For instance, I can come up with models with estimators that have zero standard deviation by sacrificing bias. Does this make these models "better"?

If my concerns above are discussed and clarified, I may be willing to raise my score.


**Questions:**

**Critique of SBNs**: Could the authors please point to where in Zhang and Matsen, (2022) I may find the discussion regarding "the limited parent-child subsplit patterns in the observed samples" (line 37 in the submission)? This is an important argument for ARTree which I have not seen investigated before. To me it could make sense to include this discussion in the Appendix of this submission, as to make the paper more stand alone.



**Limitations:**

Yes.

---

> ### Author Rebuttal · Authors · 2023-08-08
>
> Thank you for your thoughtful review and helpful feedback! We address your concerns and questions as follows.
>
> **Weakness 1**: Related work
>
> **Response**: Thanks for suggesting these related works! We will reference them and clarify the distinction between SLANTIS and ARTree in our revision. More discussions can be found in the global response.
>
> **Weakness 2**: Evaluating tree topologies
>
> **Response**: Yes, ARTree can compute $Q(\\tau')$ even if $\\tau'$ is not simulated by ARTree. It is this property that allows us to calculate KL divergence to the ground truth. This fact comes from the decomposition process in Appendix C, where we will add more detailed explanations in our revision.
>
> To calculate $Q(\\tau')$, we can sequentially remove the taxa one by one in a reversed order, starting from the last taxon being added (Algorithm 2). This way, we would get the corresponding decision sequence in linear time (Lemma 1) and use it to compute the likelihood of $\tau'$.
>
> **Weakness 3**: ''Domain expertise''
>
> **Response**: Thanks for raising this issue! In terms of domain expertise, what we indeed want to emphasize is that SBNs require a pre-selected sample of good candidate trees to provide subsplit supports for parameterization. Although running MrBayes or UFBoot does not require much domain expertise, the choice of using MCMC or bootstrapping indeed demands domain expertise (see Section 4.2 of Zhang and Matsen [2022]). Moreover, those are just some heuristic approaches that are commonly used so far, and designing efficient support estimation methods for SBNs, especially when the posterior is diffuse, remains an unsolved challenge for SBN-based VBPI.
>
> We apologize for not making this clear enough. In our revision, we will adopt your suggestion to remove the ''domain expertise'' argument for better clarification (e.g., replacing it with more specific descriptions such as pre-selected tree topologies samples).
>
> **Weakness 4**: Experiments
>
> **Response**: We are sorry that our description confused you. Let us illustrate our experiments more clearly. For each dataset, we repeat the experiment 10 times (i.e. ''trainings''). For the $i$-th repetition, we: (i) calculate the KL divergence (a deterministic number) denoted by $KL_i$; (ii) estimate the ELBO for 100 times (i.e. ''runs'') denoted by $ELBO_{i,1},\ldots, ELBO_{i,100}$, whose sample mean is $mean_{ELBO,i}$ and sample std is $std_{ELBO,i}$. We then report $mean_{KL}=\sum_i KL_i/10$, $mean_{ELBO}=\sum_i mean_{ELBO,i}/10$, and $std_{ELBO}=\sum_i std_{ELBO,i}/10$ in Table 2. The results of LB-10 and ML follow the same way as ELBO.
>
> Therefore, the stds of ELBO, LB-10, and ML across different runs reflect the variance of variational lower bounds, which is a common concern in VI. The std of KL divergence (see the following Table) across different trainings reflects the uncertainties wrt the learned model parameters, and we did not report it due to its different meaning.
>
> Table: KL divergence averaged over 10 independent trainings with standard deviation in brackets.
> |-|DS1|DS2|DS3|DS4|DS5|DS6|DS7|DS8|
> |-|---|---|---|---|---|---|---|----|
> |SBN|0.0707(0.0002)|0.0144(0.0019)|0.0554(0.0082)|0.0739(0.0012)|1.2472(0.0113)|0.3795(0.0015)|0.1531(0.0044)|0.3173(0.0257)|
> |ARTree|0.0097(0.0006)|0.0004(0.0001)|0.0064(0.0003)|0.0219(0.0014)|0.8979(0.0175)|0.2216(0.0014)|0.0123(0.0020)|0.1231(0.0078)|
>
> Finally, we want to clarify that  "100, 100, and 1000" is not a typo. In fact, we use 1000 runs for the ML estimation for a more accurate estimation of the variance.
>
> **Weakness 5**: Stds of ML results
>
> **Response**: In our experiments, the ML (in nats) was estimated with importance sampling
> $$\hat{L}_K = \log\left(\frac{1}{K}\sum\_{i=1}^{K}\frac{P(Y,q_i,\tau_i)}{Q(q_i,\tau_i)}\right)$$
> using $K=1000$ samples $(q_i,\tau_i)\sim Q(q,\tau)$, where $Q(q,\tau)$ is the variational approximation. With that many samples, the ML estimate $\hat{L}_K$ is more like an exact ML $\log p(Y)$ instead of a lower bound of it. This strategy is commonly used to access the ML of models (Normalizing Flow: http://proceedings.mlr.press/v37/rezende15.pdf; VIMCO: http://proceedings.mlr.press/v48/mnihb16.pdf). Moreover, the variance of an importance sampling estimator is often used as a measure of the approximation accuracy of the importance distribution to the target (e.g., adaptive importance sampling methods).
>
> The importance sampling estimator $\hat{L}_K$ is valid only when $Q(q,\tau)=0\Rightarrow P(Y,q,\tau)=0$. For estimators that have zero standard deviation by sacrificing bias, it seems that this condition is violated because $Q(q,\tau)$ would collapse to a point. Finally, please note that for ML in VBPI, the comparison of the stds is reasonable only when the means are in their correct range (see the following table).
>
> Table: ML estimates with std in the brackets.
> |-|DS1|DS2|DS3|DS4|DS5|DS6|DS7|DS8|
> |----|----|----|----|----|----|----|----|----|
> |ARTree|-7108.41(0.19)|-26367.71(0.07)|-33735.09(0.09)|-13329.94(0.17)|-8214.59(0.34)|-6724.37(0.46)|-37331.95(0.27)|-8650.61(0.48)|
> |MrBayes stepping-stone|-7108.42(0.18)|-26367.57(0.48)|-33735.44(0.50)|-13330.06(0.54)|-8214.51(0.28)|-6724.07(0.86)|-37332.76(2.42)|-8649.88(1.75)|
>
> **Question**: Critique of SBNs
>
> **Response**: Several relevant expressions can be found in Zhang and Matsen [2022]. For example, in the first two lines of page 9, it reads that ''$P_{\pi_i}(j\to i)$ is the conditional probability for the parent-child subsplit pair representing the local splitting pattern of ...''. Our use of the phrase ''parent-child subsplit patterns'' follows this expression. In the last 11 lines of page 11, it reads that ''if we can find a sufficiently large collection of subsplits from these favorable trees and restrict the support of CPDs accordingly ...''. This is just why we say the ''parent-child subsplit patterns'' are ''limited'' in observed samples. We will clarify this argument in Appendix A in our revised manuscript.

---

> > ### Comment · Reviewer_tL42 · 2023-08-10
> > **Response to rebuttal**
> >
> > I thank the authors for the time invested in responding to my concerns. I only have one remaining concern and one follow-up question which I expand on below. First, I would like to clarify that I do not deem it necessary to integrate the new table provided in global response into the revised version of your paper. I.e., from my point of view, you do not need to update your tables with the results from VaiPhy or VCSMC as they are barely comparable in terms of ML estimates. Regarding removing the phrase "domain expertise", I still think this is a good idea, and the modification proposed by the authors in their rebuttal (response to Weakness 3) is much more informative. Concerning Weakness 2, I must have missed this in the Appendix. Maybe you can place a clear, but brief, pointer in the main text to this part of the Appendix? I think this is a feature of ARTree that deserves amplification.
> >
> > Now, to my withstanding concern.
> >
> > **Weakness 5**
> > The generative model is parameterized with what are assumed to be known parameter values (i.e., they are not learned), meaning that the marginal log-likelihood, $\log p(y)$, is the same number regardless of the choice of variational distribution. And, indeed, as $K\rightarrow\infty$, then $L_K \rightarrow \log p(y)$, irrespective of $Q$.
> >
> > *"With that many samples, the ML estimate is more like an exact $\log p(y)$"*. I parse this statement as that you are assume that the ML estimates have zero bias from the true $log p(y)$. Aligned with what I stated above, I agree with this intuition, and the ML estimates provided in the paper are very similar in terms of their means.
> >
> > So, when evaluating the ML estimates, you are in fact interested in the qualities $Q$ has as an importance sampler? Did I understand your response correctly?
> >
> > I have to say that, if my interpretation is correct, I like this way of framing what you are testing more as the significance of achieving small std's makes more sense in the context of importance sampling. I sincerely think this should be included in the text in Sec. 4.2, as the reward of small std's right now feels ad hoc, in my opinion.
> >
> > In the context of learning $Q$'s that serve as good importance samplers, isn't it very important to train multiple $Q$'s and reason about their produced std's on average? The std's differ very little in Table 2 right now. Do you see my point? As it stands it seems like it is a bit undecisive whether SBNs or ARTrees produce the most reliable estimators?
> >
> > *Less important comments on your response*:
> >
> > First, I don't think "this strategy" is used in the NF paper by Rezende and Mohamed? The importance weighted autoencoder, and hence the new tighter objective (let's call it IWELBO), by Burda et al. was not proposed until later the same year. Also, Rezende and Mohamed report their scores as lower bounds of the negative log-likelihood (see Table 2).
> >
> > Second, just to finalize the argument about the silly estimator, the importance sampling condition you mentioned is often referred to as a rule of thumb, not a criterium for the estimator to be valid? If we choose $Q$ such that $P=0 \implies Q=0$, then $\hat{L}_K$ is still "valid" in the sense that its expectation is finite. So if I choose $Q(\tau)$ to be a categorical distribution with all its probability mass in one topology (I can only sample one topology), and $Q(q|\tau)$ to be LogNormals with super small standard deviations, my estimator of the $\log p(Y)$ would have very small standard deviation. However, as you know clarified that you consider the ML comparisons to only be reasonable when their means are in the "correct" range, this probably rules out the silly estimator here.
> >
> > **Follow-up question**
> > It is unclear to me how the KL divergence is computed to the ground truth distribution, $P$, in Table 2. Is it KL($P||Q$) or KL($Q||P$)? Since the expectations in the KLs here are taken also over continuous distributions, how do you compute these quantities? Can you evaluate a branch length sampled from $Q$ in $P$?

---

> > > ### Author Response · Authors · 2023-08-11
> > > **Thanks for your response!**
> > >
> > > Cheers for our consensus on weaknesses 1-4! Thanks for your additional suggestions! We will revise our paper accordingly. Here are our responses to weakness 5 and the follow-up question.
> > >
> > > **Response to weakness 5** Your understanding is absolutely right. We are indeed interested in the qualities $Q$ has as an importance sampler. We appreciate you agree that the significance of achieving small std's makes sense in the context of importance sampling. We will clarify this interpretation in Section 4.2, as you suggested.
> > >
> > > In the context of interpreting $Q$ as an importance sampler, we repeated VBPI 10 times on each dataset, i.e. 10 independently trained $Q$s, and reported their average std in Table 2 (please see our response to weakness 4). This way, we expect that the std estimation is more accurate. Just as you have pointed out, the stds of ML indeed differ little in Table 2. Here we provide two explanations:\
> > > (i) According to our experience, the ML (as well as ELBO and LB-10) estimates in VBPI are more sensitive to the quality of branch length model $Q(q|\tau)$ instead of the tree topology model $Q(\tau)$. As SBN and ARTree use the same branch length model, we do not expect a large improvement of (the stds of) ML.\
> > > (ii) The support of ARTree spans the entire tree topology space. This adds to the difficulty of training $Q(q|\tau)$ which is conditioned on tree topology $\tau$, as discussed in Appendix E.
> > >
> > > Therefore, we only expect ARTree to be comparable to SBNs in terms of ML: this indicates that ARTree works well together with a collaborative branch length model for VBPI. The strong power of ARTree for modeling tree topologies is mainly reflected by the KL results (please see our response to the follow-up question for more details).
> > >
> > > *About the ''less important comments''*
> > >
> > > First. We agree that the interpretation of ML results in Table 2 in the NF paper is different from our paper and they did not use multi-sample lower bound for training. What we wanted to express in our response is that the idea of importance sampling is used in the NF paper to estimate ML (page 7: ''The true marginal likelihood is estimated by importance sampling using 200 samples from the inference network''). As far as we know, although the IWELBO was later proposed as an optimization objective, the idea of importance sampling for marginal likelihood evaluation, or more generally numerical integration, has long existed.
> > >
> > > Second. We apologize that we probably had a different understanding of ''validness''. In our response, we said $\hat{L}_K$ is valid in the sense that it is a strongly consistent estimator of $\log p(Y)$ as $K\to\infty$. This requires $Q=0\Rightarrow P=0$. We think it's just a different usage of this word. Finally, thank you for providing this interesting example!
> > >
> > > **Response to the follow-up question** We are sorry for the confusion. The KL divergence in our paper is $KL(P(\tau|Y)|Q(\tau))$ instead of $KL(P(q,\tau|Y)|Q(q,\tau))$, i.e. the approximation accuracy of the **marginal distribution of tree topology** (please see line 228 and line 238-240). Therefore, the strong power of ARTree for modeling tree topologies is reflected by the significantly improved KL results over SBNs.

---

> > > > ### Comment · Reviewer_tL42 · 2023-08-15
> > > > **Thanks!**
> > > >
> > > > Thanks for the nice discussion. I think the submission, and the VI-based phylogeny field in general, would benefit substantially by included a revised version of our discussion above regarding the connection between the ML estimates and the evaluations of $Q$ as an importance sampler. Please include this in the final version of the paper, as you have stated.
> > > >
> > > > I will increase my score from 6 to 7.
> > > >
> > > > **Regarding KL**
> > > > Ok, now I understand what you are measuring. And so the KL scores for ARTree in Table 1 and Table 2 are different as in Table 2 the scores were produced when learning $Q(\tau)$ using VBPI and in Table 1 only Eq. 13 was used?
> > > >
> > > > This is very interesting, as it implies that the maximum likelihood learning objective in Eq. 13 gives better tree-topology posterior approximations than the VI approach (smaller KL). Do you suspect that this would change if you instead trained ARTree using $K=1$ when learning $Q$ using the ELBO? Maybe this should be highlighted in the conclusion section?

---

> > > > > ### Author Response · Authors · 2023-08-17
> > > > > **Thanks**
> > > > >
> > > > > Thank you for raising the score! We will include the discussion in our revision.
> > > > >
> > > > > **Regard KL** Thanks for this question. The KL results of ARTree in Table 1\&2 differ, as the results in Table 1 are obtained using maximum likelihood estimation (MLE) while the results in Table 2 are obtained using variational Bayesian phylogenetic Inference (VBPI). Generally, MLE is easier for ARTree because it is trained with the high-quality tree topology samples directly; in VBPI, ARTree has to explore the entire tree topology space and collaborates with a branch length model to fit the joint distribution $P(\tau,q|Y)$. Therefore, the KL results of MLE are expected to be better than those of VBPI.
> > > > >
> > > > > We are not sure if a smaller $K$ would improve the approximation quality of $Q(\tau)$. Note that in our current approach, we use multi-sample lower bound $K>1$ for efficient stochastic gradient estimator of tree topology parameters (e.g., VIMCO). These gradient estimators require multi-sample objectives ($K>1$). Moreover, large $K$ would also encourage exploration over the vast and multimodal tree space. However, larger $K$ would also decrease the signal-to-noise ratio [1] and hence make it harder to obtain accurate variational approximations. We will highlight this in the conclusion section in our revision.
> > > > >
> > > > > [1] Rainforth, Tom, et al. "Tighter variational bounds are not necessarily better." ICML, 2018.

---

### Official Review · Reviewer_vFMY · 2023-07-06

**Soundness:** 4 excellent
**Presentation:** 4 excellent
**Contribution:** 4 excellent
**Rating:** 8
**Confidence:** 4

**Summary:**

The paper introduces ARTree, a deep generative autoregressive model for phylogenetic tree reconstruction. The authors define an autoregressive sequential process for generating tree topologies, $\tau$, and prove that there is a one-to-one mapping between the resulting topologies and the decision sequence, $D$, instantiated by the process. Utilizing this fact, the authors define a distribution over $D$, letting each decision at time n be drawn from a Categorical distribution given previous decisions 1,..., n-1, and parameterize these Categoricals by calculating and passing learnable topological features to Graph Neural networks (GNN) with a recurrent unit and unit to incorporate time embeddings.
ARTree is then used as variational distribution for tree topologies $Q(\tau)$, along with a GNN based parameterization of the branch length distribution, within Variational Bayesian phylogenetic inference (VBPI). The representative power of $Q(\tau)$ is evaluated by comparing it with Subsplit Bayesian networks (SBNs) to a "ground truth" posterior distribution (based on the posteriors of long-running MrBayes experiments) and within the context of VBPI on benchmark datasets.

**Strengths:**

The vastness of the tree topology space in phylogenetic inference is a well-known obstacle in both classical and tumor phylogenetics. The VI approach to Bayesian phylogenetics is an ongoing area of research and is need of more sophisticated tree topology variational distributions and experiment design to evaluate these varaiational distributions. The tree topology density experiment in 4.1 not only shows strong representative power of ARTree, but the experiment design itself is a contribution to the research field. Furthermore, ARTree is a large improvement when compared to previous VI-methods in phylogeny that use unconfined $Q(\tau)$ (however this is not highlighted in the paper, see Weaknesses section).

**Weaknesses:**

The paper fails to mention other works within VI in phylogenetics, e.g., VaiPhy (https://arxiv.org/abs/2203.01121) and VCSMC (https://arxiv.org/abs/2106.00075); these methods do not confine the $Q(\tau)$ support either and are relevant related work.

The paper fails to highlight the strong performance of ARTree in VBPI w.r.t. other VI methods with unconfined $Q(\tau)$ - adding a row to Table 2 with results from, e.g., VCSMC and VaiPhy would greatly accentuate the contribution of the paper.

ARTree relies on several subroutines to be able to construct and parameterize the generative decision sequences. This makes the method complicated to grasp and implement, which, given the complex problem at hand, can be regarded as a strength in the authors perseverance rather than a weakness of the paper. However, the amount of steps involved, e.g., L steps of messing passing and calculating the topological node embeddings, naturally invokes questions regarding inference runtime and memory usage. The lack of runtime comparisons between VBPI with ARTree, other methods in VI and MrBayes is a weakness of the paper.

**Questions:**

Overall, a strong contribution to field of Bayesian phylogenetics and a well-written paper.

The addressing the following points could make me increase my score further:
1. Experiment on runtime of ARTree in the context of VBPI
2. Addressing the issue of minimal increase of raised in Weaknesses

Failing to incorporate the related works mentioned in Weaknesses-section could make me decrease my score.

Misprints:
56 "edges of current…" -> "edges of the current…"

**Limitations:**

Limitations, except for potential runtime concerns (see Weaknesses-section), are properly addressed.

---

> ### Author Rebuttal · Authors · 2023-08-08
>
> Thank you for your careful review and valuable feedback! Below are our answers to your comments:
>
> **Weakness 1**: The paper fails to mention other works within VI in phylogenetics, e.g., VaiPhy and VCSMC; these methods do not confine the $Q(\tau)$ support either and are relevant related work.
>
> **Response**: Thank you for pointing out these missing related works. We will include the comparison between ARTree and the two important related contributions, VaiPhy[1] and VCSMC[2], in our revision. Please see our global response for more discussions.
>
> **Weakness 2**: The paper fails to highlight the strong performance of ARTree in VBPI w.r.t. other VI methods (...) greatly accentuate the contribution of the paper.
>
> **Response**: Thanks for the suggestion! We will add a comparison between ARTree and other VI methods, e.g. VaiPhy and VCSMC (see the following table) in Table 2 in our revised manuscript. We did not include this in our original paper considering SBN as the SOTA model outperforms other methods significantly.
>
> Table:  Marginal likelihood (ML) estimates with one standard deviation in the brankets. $\phi$-CSMC is proposed along with VaiPhy in [1] and works on bifurcating tree topology space, making it comparable with other methods.
>
> | ---- | ARTree | SBN | VCSMC[2] | $\phi$-CSMC[1] |
> | ---- | ---- | ---- | ---- | ---- |
> | DS1 | -7108.41(0.19) | **-7108.41(0.15)** | -9180.34(170.27) | -7290.36(7.23) |
> | DS2 | **-26367.71(0.07)** | -26367.71(0.08) | -28700.7(4892.67) | -30568.49(31.34) |
> | DS3 | **-33735.09(0.09)** | **-33735.09(0.09)** | -37211.20(397.97) | -33798.06(6.62) |
> | DS4 | **-13329.94(0.17)** | -13329.94(0.20) | -17106.10(362.74) | -13582.24(35.08) |
> | DS5 | **-8214.59(0.34)** | -8214.62(0.40)| -9449.65(2578.58) | -8367.51(8.87) |
> | DS6 | -6724.37(0.46) | **-6724.37(0.43)** | -9296.66(2046.70) | -7013.83(16.99) |
> | DS7 | **-37331.95(0.27)** | -37331.97(0.28) | N/A | N/A |
> | DS8 | **-8650.61(0.48)** | -8650.64(0.50) | N/A | -9209.18(18.03) |
>
> [1] Koptagel, Hazal, et al. "VaiPhy: a Variational Inference Based Algorithm for Phylogeny." NeurIPS 2022.\
> [2] Moretti, Antonio Khalil, et al. "Variational combinatorial sequential Monte Carlo methods for Bayesian phylogenetic inference." UAI 2021.
>
> **Weakness 3**: ARTree relies on several subroutines to be able to construct and parameterize the generative decision sequences. (...) The lack of runtime comparisons between VBPI with ARTree, other methods in VI and MrBayes is a weakness of the paper.
>
> **Response**: Thank you for pointing out the lack of runtime comparisons. We will add runtime comparisons in Appendix E in the revised manuscript. The following table is the CPU time and memory of each method in the VI setting on DS1. We do not compare them with the MCMC-based MrBayes because it seems hard to determine a fair time criterion as MrBayes is written in C++ and VBPI is written in Python.
>
> Table: The CPU time and memory usage in the VI setting on DS1. The CPU time is averaged over 100 trials with one standard deviation in the brackets. The experiments are run on a single core of MacBook Pro 2019. N/A: not available due to unresolved memory leak issues.
> | ---- |  SBN   | ARTree  | VCSMC | VaiPhy |
> | ---- |  ----  | ----  | ----  | ----  |
> | CPU time of passing 100 trees (seconds) | 0.99(0.14)  | 5.61(0.22) | 11.54(1.50)  | 34.97(0.63)  |
> | Memory (MB) | 611.74  | 605.78 | N/A  | N/A |
>
> Although VCSMC and VaiPhy seem to take longer time if the number of trees is fixed, they generally require hundreds of iterations to converge, since their variational distributions only have a few parameters and are highly structured. In contrast, SBN and ARTree require more than 10w iterations, since they both build machine-learning models with enormous parameters and rely heavily on optimization. ARTree takes more time than SBN because it relies on several submodules which, although complicated, are designed to promote the expressive power to accommodate the complex tree space and are widely-used strategies in the literature. The inefficiency of autoregressive generative models is also an inherent issue.
>
> The following strategies may help to reduce the computational cost of ARTree. (i) Training on GPUs. As a deep model, ARTree is mainly implemented using vectorized tensor operations in PyTorch. (ii) Early stopping. Although ARTree is trained for 40w iterations in VBPI to get the best numerical results, 10w iterations are enough to reveal the ground truth trees. (iii) More efficient architecture. Several efforts have been made to accelerate autoregressive models, e.g. GraphGEN (https://arxiv.org/abs/2001.08184). Designing efficient architectures for ARTree is an important future direction.
>
> **Question 1**: Experiment on runtime of ARTree in the context of VBPI.
>
> **Response**: Please see our response to weakness 3.
>
> **Question 2**: Addressing the issue of minimal increase of raised in Weaknesses.
>
> **Response**: We are sorry that we could not understand this question. We could not find relevant expressions about 'minimal increase' in Weaknesses.
>
> **Question 3**: Incorporating the related works mentioned in Weaknesses-section.
>
> **Response**: Please see our response to weaknesses 1 and 2.
>
> **Question 4**: Misprints: 56 "edges of current…" -> "edges of the current…".
>
> **Response**: Thank you for your careful check. We will fix this misprint in our revision.

---

> > ### Comment · Reviewer_vFMY · 2023-08-16
> > **Rebuttal response**
> >
> > Thank you for addressing the concerns raised in my review. The suggested updates in the global response and to weakness 1 and 2 will serve the paper well. However, I still have concerns regarding Weakness 3 and clarify question 2 below.
> >
> > Weakness 3:
> > The added experiment on time to pass topologies is interesting, however, the issue I raised was regarding runtime of ARTree for VI, supposed to show the runtime needed to produce the results of Table 2. This way readers can see the current trade-off between performance and runtime for different VI approaches. The extension of Table 2 together with the CPU table added conceals this trade-off in learning time and performance, which is very misleading.
> >
> > I find the argument regarding runtime comparisons to MrBayes fair.
> >
> > Question 2:
> > My apologies, the full formulation of that question must have been lost in one of my offline sessions. It should have been: What is the reason behind the seemingly minimal increase of ELBO between ARTree and VPBI? Does this come from a different $q(B | T)$ or does it in fact come from the different $q(T)$? Maybe this is hard to disentangle, but addressing this fact would be interesting for the discussion.
> >
> >
> > Currently, I will retain my score as the rebuttal did not address weakness 3 in a satisfactory manner.

---

> > > ### Author Response · Authors · 2023-08-18
> > > **Thanks for your response!**
> > >
> > > Thanks for your response! We addressed weakness 3 and question 2 as follows.
> > >
> > > **Response to weakness 3** We apologize for misunderstanding your concern. Here is the runtime comparison of different methods in the VI setting on DS1 (will be added to the limitation section).
> > >
> > > Table: Runtime comparison in the VI setting on DS1. SBN* and ARTree* refer to the early stopping of SBN and ARTree that surpass the $\phi$-CSMC baseline in terms of marginal likelihood estimation (-7290.36), respectively.
> > > | ---- |   VCSMC | VaiPhy |  $\phi$-CSMC | SBN   | ARTree | SBN* | ARTree* |
> > > | ---- |  ----  | ----  | ----  | ----  | ----  | ----  | ----  |
> > > | Total training time (minutes) | 248.3 | 45.1 | N/A | 659.3 | 3740.8 | 10.2 | 79.5 |
> > > | Evaluation time (one estimate of ML, minutes) | 2.4 | 1.6 | 102.2 | 0.15 | 0.41 | 0.15 | 0.41 |
> > >
> > > *Remarks of the table*: (i) **Training**. We trained all models following the setting in their original papers: VCSMC was trained for 100 iterations with 2048 particles per iteration; VaiPhy was trained for 200 iterations with 128 particles per iteration; $\phi$-CSMC directly estimates ML based on VaiPhy, and therefore does not need extra training; both ARTree and SBN were trained for 400000 iterations with 10 particles per iteration. (ii) **Evaluation**. We find that the evaluation strategies in their original papers are quite different, e.g. VaiPhy, SBN, and ARTree used importance sampling to estimate ML with different repetition times; VCSMC and $\phi$-CSMC instead estimated ML with sequential Monte Carlo (SMC), also with different repetition times. To be fair, we report the time for producing one estimate of ML from each of these models (VaiPhy, SBN, and ARTree used importance sampling with 1000 particles; VCSMC and $\phi$-CSMC used SMC with 2048 particles).
> > >
> > > We want to emphasize that although ARTree (and SBN) takes longer time to converge (complete training) when compared to other methods with unconfined support, it takes a comparable amount of time to provide good enough approximations for marginal likelihood estimation of similar accuracy (see ARTree* and SBN* on the above Table). Moreover, the evaluation time of ARTree (and SBN) for marginal likelihood estimation would be much shorter than other methods.
> > >
> > > The suggestions for reducing the computational costs in our rebuttal are still applicable, among which designing a more efficient architecture for ARTree is an important future direction.
> > >
> > > **Response to question 2** Thanks for this question. Just as you have pointed out, the improvement of ELBO in Table 2 is minor. Here we provide two explanations: (i) The ELBO estimates in VBPI are more sensitive to the quality of branch length model $Q(q|\tau)$ instead of the tree topology model $Q(\tau)$. As SBN and ARTree use the same parametrization of the branch length model, we do not expect a large improvement in ELBO. (ii) The support of ARTree spans the entire tree topology space. This adds to the difficulty of training $Q(q|\tau)$ which is conditioned on tree topology $\tau$, as discussed in Appendix E.
> > >
> > > To investigate whether the increase of ELBO comes from a different $Q(q|\tau)$ or a different $Q(\tau)$, we conducted the following experiment (see the table).
> > >
> > > Table: The ELBO estimates on DS1 obtained by different combinations of tree topology model $Q(\tau)$ and branch length model $Q(q|\tau)$.
> > > | Model combination | ELBO |
> > > | ---- | ---- |
> > > | SBN + branch length model trained along with SBN | -7110.24(0.03) |
> > > | SBN + branch length model trained along with ARTree | -7110.26(0.03) |
> > > | ARTree + branch length model trained along with ARTree | -7110.09(0.04) |
> > >
> > > Therefore, it seems that the increase of ELBO indeed comes from a different $Q(\tau)$, as evidenced by the result of the 'SBN + branch length model trained along with ARTree' combination. This observation also coincides with the explanation (ii).

---

> > > > ### Comment · Reviewer_vFMY · 2023-08-21
> > > > **All concerns addressed**
> > > >
> > > > Weakness 3
> > > > This added analysis greatly improves the paper as it fully shows the current state of different VI approaches to phylogeny, especially with the added ARTree* and SBN*, in terms of performance and runtime. I will raise my score further to an 8.
> > > >
> > > > Question 2
> > > > Thank you for adding additional experiments and an interesting discussion for this question. This does suggest future work in exploring more complicated data sets where the different $Q(\tau)$ support could be even more divergent, which could be relevant for downstream tasks. I think this added experiment and analysis would be a good contribution to the paper, although including it or not will not effect my score.
> > > >
> > > >
> > > > Thank you for a interesting and well-written paper, rebuttal and discussion.

---

> > > > > ### Author Response · Authors · 2023-08-21
> > > > > **Thanks!**
> > > > >
> > > > > Thanks for your valuable suggestions again! We will include our discussions about runtime and ELBOs in the revised version. These will indeed make our paper stronger :)

---

### Official Review · Reviewer_aQBU · 2023-07-19

**Soundness:** 4 excellent
**Presentation:** 2 fair
**Contribution:** 3 good
**Rating:** 7
**Confidence:** 4

**Summary:**

In this paper, the authors propose a tractable distribution over tree spaces that can be fit for use in density estimation or variational inference.  The key idea is to build a tree by sequentially adding leaves one at a time by adding an additional branch to the tree.  Each step of this process has a reasonable state space, and it is easy to see that such a process generates distributions that span all of tree space.  The authors then parameterize the "action space" of this process using (recurrent) graph neural networks, which can be trained using either maximum likelihood in the density estimation case or VIMCO to optimize the ELBO in the VI case.  The authors apply their method to 8 standard phylogenetic benchmarking datasets finding comparable or superior performance to existing density estimation .or VI methods.

**Strengths:**

* The ideas presented in this paper are extremely simple and elegant.
* The big picture of the approach is easy to describe and conceptually straightforward.
* The performance of the method seems to be an advance over existing methods, even by metrics that favor existing methods (e.g., inclusive KL is kind to SBNs, as exclusive KL would be infinite for SBNs that do not have support on all of tree space).

**Weaknesses:**

* A more thorough description of the technical details of the parameterization of the model would be helpful. In particular, it would be useful to have a schematic representing all of the components and how they fit together. Equations 7-11 had a lot of subcomponents (e.g., $P$ and $R$ and $b_n$ and emb, etc... etc...) which were hard to keep track of and see how they all fit together.


Minor:
* I know that it is common in the field, but it is not obvious to me why one would want to take $K$ greater than $1$ in equation (4).  If $K$ is $1$, then (4) is exactly the usual ELBO.  Maximizing the $K=1$ ELBO corresponds to minimizing the KL between the variational and true posteriors, which seems desirable.  Taking $K$ larger than one certainly tightens the lower bound on the evidence, but that doesn't necessarily mean that it will result in a better variational approximation to the posterior.  In fact, as $K \to \infty$ equation (4) should become independent of $Q$, which seems undesirable.  See for example https://proceedings.mlr.press/v80/rainforth18b.html
* Many of the references at the end have minor formatting issues (e.g., lacking capitalization: "bayesian,  "markov", "monte carlo", "Graphrnn", etc...)

**Questions:**

Have the authors explored how necessary it is to condition the decisions on all previous decisions?  Does a Markov decision process perform substantially worse?  That is, does one need the recurrent GNN, or would a non-recurrent GNN be sufficient?

**Limitations:**

The authors have adequately addressed the limitations of their study, and I do not foresee any potential negative societal impacts.

---

> ### Author Rebuttal · Authors · 2023-08-08
>
> Thank you for your helpful feedbacks and suggestions! Here are our responses to them.
>
> **Weakness 1**: A more thorough description of the technical details of the parameterization of the model would be helpful. In particular, it would be useful to have a schematic representing all of the components and how they fit together. Equations 7-11 had a lot of subcomponents (e.g., $P$ and $R$ and $b_n$ and emb, etc... etc...) which were hard to keep track of and see how they all fit together.
>
> **Response**: Thanks for the suggestion! We will modify our description and notations accordingly in our revision to make it more clear to the readers.
>
> **Weakness 2**: I know that it is common in the field, but it is not obvious to me why one would want to take $K$ greater than $1$ in equation (4). (...) See for example https://proceedings.mlr.press/v80/rainforth18b.html.
>
> **Response**: Thanks for asking! There are mainly two reasons for taking $K>1$. (i) The gradient of variational bound w.r.t. the discrete component $\tau$ is generally unstable and suffers from large variance. Taking $K>1$ allows us to use efficient stochastic gradient estimators such as VIMCO (which are designed for multi-sample ELBO) for the tree topology variational parameters. (ii) A sample size $K$ larger than $1$ may encourage exploration over the vast and multimodal tree space to avoid being trapped in local modes.
>
> We agree that taking $K$ larger does not necessarily lead to a better variational approximation, as the signal-to-noise ratio decreases as $K$ increases (https://proceedings.mlr.press/v80/rainforth18b.html). In practice, a moderate $K$ would be good choice, and we leave a more thorough investigation on the effect of $K$ to future work. Thanks for bringing up this discussion, and we will cite this paper in our revision.
>
> **Weakness 3**: Many of the references at the end have minor formatting issues (e.g., lacking capitalization: "bayesian, "markov", "monte carlo", "Graphrnn", etc...)
>
> **Response**: We appreciate your careful review of the references. All the formatting issues will be carefully addressed by us in the revised version of the paper.
>
> **Question**: Have the authors explored how necessary it is to condition the decisions on all previous decisions? Does a Markov decision process perform substantially worse? That is, does one need the recurrent GNN, or would a non-recurrent GNN be sufficient?
>
> **Response**: Thanks for your insightful question! We haven't explored the option that uses Markov decision process. However, we expect it to work fairly well given that the current tree topology is a summarization of all previous decisions.

---

> > ### Comment · Reviewer_aQBU · 2023-08-10
> >
> > Thank you for the clear response!

---

> > > ### Author Response · Authors · 2023-08-11
> > >
> > > Thanks for your careful review and helpful suggestions again!

---

### Official Review · Reviewer_tXZx · 2023-07-21

**Soundness:** 3 good
**Presentation:** 3 good
**Contribution:** 2 fair
**Rating:** 6
**Confidence:** 3

**Summary:**

This paper presents a new way to construct the variational distribution of tree topologies based on autoregressive generation with GNNs, which is used in the problem of variational phylogenetic inference. The paper mainly compares the new construction of the variational distribution with subsplit Bayesian network approaches. The experiments show that the proposed approach outperforms SBN in terms of learning the ground-truth tree topologies.

**Strengths:**

1. The paper seems to address an interesting sub-problem of the task, which is how to flexibly generate tree topologies for the variational distribution. I'm not a domain expert in variational phylogenetic inference, but previous approaches usually follow the way of SBNs and this paper proposes a novel and better alternative to SBNs.

2. The proposed approach of autoregressive generation with GNNs looks intuitive. The paper provides comprehensive experiments in the comparison with SBN, which support the claims of the proposed method.

3. The paper is well-written and easy to follow.

**Weaknesses:**

1. As stated in the paper, the main drawback of SBN is that it could not span the entire tree topology space. It seems that there is no analysis on how/why the proposed method is better at doing this in addition to the empirical comparison in the experiments.

2. The main contribution of the paper is in Section 3.1, which is a new parameterisation of $Q(r)$. Most of the techniques in Section 3.2 follows Zhang (2023). It might be better to shorten 3.2, as I think it might not the focus of the paper.

3. Although Table 1 shows that ARTree has better number in terms of revealing the ground-truth trees, it seems that ARTree does not improve much on ELBO and ML in Table 2.

**Questions:**

Can the authors talk about the potential use of the proposed method in computational biology in addition to getting better EBLO or other metrics of modelling the data? such as interpretability.

---

> ### Author Rebuttal · Authors · 2023-08-08
>
> Thank you for your careful review and valuable questions! We address your comments and questions as below.
>
> **Weakness 1**: As stated in the paper, the main drawback of SBN is that it could not span the entire tree topology space. It seems that there is no analysis on how/why the proposed method is better at doing this in addition to the empirical comparison in the experiments.
>
> **Response**:  Thanks for your question! In the generating process of ARTree, we use the softmax function to parameterize the conditional probability of decisions (where to add the new tip node). Therefore, all possible decisions would have nonzero probabilities. This means ARTree can sample any decision sequence with a nonzero probability. As there is a bijection between decision sequences and the entire tree topology space (Theorem 1), this implies ARTree can span the entire tree topology space. We will make it more clear in our revision.
>
> **Weakness 2**: The main contribution of the paper is in Section 3.1, which is a new parameterisation of $Q(\tau)$. Most of the techniques in Section 3.2 follows Zhang (2023). It might be better to shorten 3.2, as I think it might not the focus of the paper.
>
> **Response**: Thanks for the suggestion! We will modify it accordingly in our revision.
>
> **Weakness 3**: Although Table 1 shows that ARTree has better number in terms of revealing the ground-truth trees, it seems that ARTree does not improve much on ELBO and ML in Table 2.
>
> **Response**: Yes, you are right! In fact, the power of ARTree for VBPI is mainly on tree topology approximation as reflected by the KL results in Table 2. There are two reasons for minor improvements on lower bounds. (i) According to our experience, the lower bounds in VBPI are more sensitive to the quality of branch length model $Q_\psi(q|\tau)$ than to the tree topology model $Q_\psi(\tau)$, and ARTree and SBN use the same branch length model in VBPI. Also, we want to clarify that significantly improving ELBO and LB-10 are difficult considering they approach the same marginal likelihood. (ii) The support of ARTree spans the entire tree topology space. This adds to the difficulty of training $Q_\psi(q|\tau)$ which is conditioned on tree topology $\tau$, as discussed in Appendix E.
>
> **Question**: Can the authors talk about the potential use of the proposed method in computational biology in addition to getting better ELBO or other metrics of modelling the data? such as interpretability.
>
> **Response**: This is an interesting and open question. Two potential uses: (i) ARTree provides an alternative family of distributions over the entire tree topology with explicit likelihood computation and flexibility. This is itself a useful tool for phylogenetic inference, including tree density estimation and variational posterior approximations, which has a wide range of applications such as genomic epidemiology and conservation genetics. (ii) In ARTree, these learned conditional distributions for the species attaching operations also carry important information about the relationship between the new species to the species on the current tree topologies, and hence can be used to interpret the closeness among these species.

---

> > ### Comment · Reviewer_tXZx · 2023-08-19
> > **Thanks for the response**
> >
> > I've read the authors' response and other reviewers' comments. I will keep my score.

---

> > > ### Author Response · Authors · 2023-08-21
> > > **Thanks!**
> > >
> > > Thank you for your careful review and valuable questions again!

---

### Author Rebuttal · Authors · 2023-08-08

We thank all reviewers for their constructive feedback.  We have incorporated their suggestions and will revise the paper with the following major changes:

**Related works**: We will clarify the distinction between ARTree and two related works - VaiPhy[1] and VCSMC[2] (see a short discussion below). Experimental comparisons will also be added to Table 2.

 - **A short comparision**. Both VaiPhy[1] and VCSMC[2] have unconfined support over tree topology spaces. VaiPhy employs a novel sequential sampler named SLANTIS, which makes decisions on adding edges in a specific order to sample multifurcating tree topologies. Unlike ARTree which uses parametrized GNNs, SLANTIS derives decisions based on a simply parameterized weight matrix and maximum spanning trees. Secondly, VCSMC samples tree topologies through subtree merging and resampling following CSMC[3], but employs a parametrized proposal distribution. A more powerful variant of it called variational nested SMC (VNCSMC) gives better proposals by incoporating future iterations. In contrast, ARTree takes a different approach by employing GNNs for an autoregressive model that builds up the tree topology sequentially, without requiring a resampling step or a looking-forward step. Empirically, we find ARTree surpasses these two methods significantly in terms of marginal likelihood.

Table:  Marginal likelihood (ML) estimates with one standard deviation in the brankets. $\phi$-CSMC is proposed along with VaiPhy in [1] and works on bifurcating tree topology space, making it comparable with other methods.
| ---- | ARTree | SBN | VCSMC[2] | $\phi$-CSMC[1] |
| ---- | ---- | ---- | ---- | ---- |
| DS1 | -7108.41(0.19) | **-7108.41(0.15)** | -9180.34(170.27) | -7290.36(7.23) |
| DS2 | **-26367.71(0.07)** | -26367.71(0.08) | -28700.7(4892.67) | -30568.49(31.34) |
| DS3 | **-33735.09(0.09)** | **-33735.09(0.09)** | -37211.20(397.97) | -33798.06(6.62) |
| DS4 | **-13329.94(0.17)** | -13329.94(0.20) | -17106.10(362.74) | -13582.24(35.08) |
| DS5 | **-8214.59(0.34)** | -8214.62(0.40)| -9449.65(2578.58) | -8367.51(8.87) |
| DS6 | -6724.37(0.46) | **-6724.37(0.43)** | -9296.66(2046.70) | -7013.83(16.99) |
| DS7 | **-37331.95(0.27)** | -37331.97(0.28) | N/A | N/A |
| DS8 | **-8650.61(0.48)** | -8650.64(0.50) | N/A | -9209.18(18.03) |

[1] Koptagel, Hazal, et al. "VaiPhy: a Variational Inference Based Algorithm for Phylogeny." NeurIPS 2022.\
[2] Moretti, Antonio Khalil, et al. "Variational combinatorial sequential Monte Carlo methods for Bayesian phylogenetic inference." UAI 2021.\
[3] Wang, Liangliang, Alexandre Bouchard-Côté, and Arnaud Doucet. "Bayesian phylogenetic inference using a combinatorial sequential Monte Carlo method." Journal of the American Statistical Association (2015).

**Technical details**: We will provide a more schematic description of the technical details in Section 3.2 and remove redundant statements to present it more clearly to the readers.

**Limitations**: We will add a runtime comparison in Appendix E and give some suggestions for reducing the computational cost.

**Decomposition process**: We will add a description of how to evaluate the tree topology probability using the decomposition process in Appendix C.

**Drawback of SBNs**: We will remove the ambiguous argument about ''domain expertise'' and emphasize the importance of high-quality pre-sampled trees. We will clarify why SBN faces ''the limited parent-child subsplit patterns in the observed samples'' in Appendix A.

We hope our response has adequately addressed the reviewers' questions and concerns, and look forward to reading any other additional comments.

---

### Decision · Program_Chairs · 2023-09-21

**Decision:**

Accept (spotlight)

**Comment:**

This paper presents a new way to construct the variational distribution of tree topologies based on autoregressive generation with GNNs, which is used in the problem of variational phylogenetic inference.

The authors have done a pretty good job of rebuttal, and after rebuttal, this paper received scores of 6778. Most of the reviewers are confident that this paper should be accepted. Specifically, reviewers have commented that this submission is a strong contribution to the field of Bayesian phylogenetics and a well-written paper. The ideas presented are simple and elegant, and comprehensive experiments have been done in the comparison with SBN, which support the claims of the proposed method. Overall, given the general positive review feedback, the AC would like to recommend acceptance of the paper.